# Low-dimensional topology of deep neural networks

**Junyu Ren** [1]   **Lek-Heng Lim** [1]

## Abstract

We study layered models, including feedforward networks, ResNets, and transformers, by limiting each layer to a width of $d = 3$, i.e., $\mathbb{R}^3$ as representation space. This allows us to track how a neural network changes low-dimensional topological invariants through its layers. Just about any topological structure may be simplified or even trivialized by simply increasing dimension; e.g., any knot is equivalent to an unknot in $\mathbb{R}^4$. By restricting to $\mathbb{R}^3$, we not only isolate the effects of activation and depth from that of width, we work in a space that lends itself to easy visualization. We focus on *linking number* here, deferring other invariants like link groups, Milnor's $\bar{\mu}$-invariants, knot types, ambient cobordisms, to a sequel. We provide full proofs and empirical experiments to justify the following insights: When measured by their power to effect changes in linking numbers, the layer-skipping feature in ResNets is as powerful as the attention mechanism in transformers; both ResNets and transformers are strictly more powerful than feedforward neural networks with monotonic activations, which are in turn more powerful than invertible and flow-based models; but replacing monotonic activation with a nonmonotonic one elevates a feedforward network into the same expressivity class as ResNets and transformers. These results suggest that low-dimensional topology can be a useful tool to guide designs of AI architectures. We also generalize our results from $d = 3$ to arbitrary $d > 3$.

## 1. Introduction

The last 50 years in topology has arguably been the half-century of *low-dimensional topology*, with many astounding groundbreaking advances in the topology of 3-manifolds

[1]University of Chicago, Chicago, IL, USA. Correspondence to: Junyu Ren <junyuren@uchicago.edu>.

*Proceedings of the 43$^{rd}$ International Conference on Machine Learning*, Seoul, South Korea. PMLR 306, 2026. Copyright 2026 by the author(s).

(Wall, 1984; Birman, 1991; Lott, 2007) and 4-manifolds (Atiyah, 1987; Milnor, 1987). We will show how innovative topological invariants developed to study low-dimensional (i.e., 3- or 4-dimensional) manifolds may be put to good use in the analysis and design of modern AI models.

We begin with a simple example. Consider two interlocking rings in $\mathbb{R}^3$, the Hopf link. Can a neural network learn to classify points on these rings into their respective connected components? This deceptively simple question reveals deep connections between topology and machine learning. Under the manifold hypothesis (Bengio et al., 2013), real-world data concentrates near low-dimensional manifolds embedded in high-dimensional space. When class manifolds are topologically entangled, like in a Hopf link, classification *requires* one to geometrically "untangle" them. We will see how such topological obstructions impose fundamental constraints on neural network architectures. Due to space constraints, we limit ourselves to one specific invariant, the linking number, to illustrate our general framework of using low-dimensional topological invariants to study neural network architectures. A companion paper treats more sophisticated invariants in detail (Ren & Lim, 2026).

We emphasize that dimension itself is a powerful, if not all-powerful, attribute in machine learning (Cover, 1965; Cortes & Vapnik, 1995) — there is almost nothing that one cannot do through simply increasing dimension, including effecting topological changes; although curse-of-dimensionality often also accompanies a dimension increase. The flip side of this coin is that dimension is a factor that masks the effects of all other aspects of a neural network architecture. Our approach removes this confounding factor by fixing width throughout, first at $d = 3$, and later increasing it to higher $d$.

Existing neural network theory largely ignores topological structure other than dimension (a topological invariant). Universal approximation theorems (Cybenko, 1989; Hornik et al., 1989) show that sufficiently wide networks approximate any function, but say nothing about which architectures succeed on which data. Depth–width tradeoffs (Telgarsky, 2016; Eldan & Shamir, 2016) and approximation rates (Barron, 1993; Yarotsky, 2017) characterize function complexity, not data complexity. Topological data analysis (Carlsson, 2009) studies intrinsic data shape but not how embeddings in ambient space constrain learning. Meanwhile, practi-

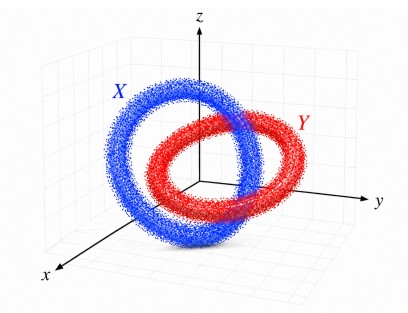 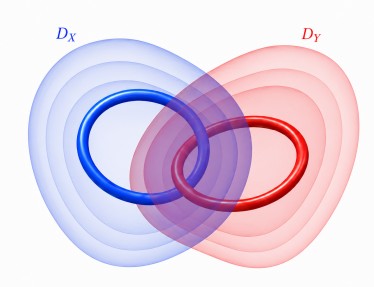 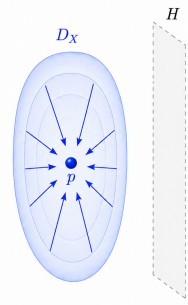 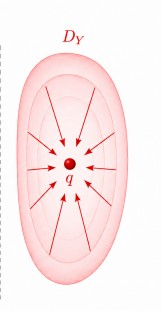

*(a)* point cloud data with linked supports

*(b)* convex decision regions containing linked classes must intersect

*(c)* classes separated by a hyperplane cannot be linked

*Figure 1.* Linked supports create a topological obstruction to classification: linked class manifolds cannot be contained in disjoint convex decision regions, while hyperplane-separated classes are necessarily unlinked.

tioners have observed that certain architectural features like skip connections, nonmonotonic activations, attention, etc, consistently outperform alternatives. Engineering reasons have been proffered for some of these: training stability, information flow, incremental layer-wise transformations, etc. We will provide a topological perspective.

When does topological structure create fundamental barriers for neural networks, and which mechanisms overcome them? The Hopf link (Figure 1) provides a canonical example: two interlocking circles with linking number $\text{link} = 1$, measuring how many times one curve winds through the other. Linear separability requires $\text{link} = 0$. Indeed, if a hyperplane separates two curves, each curve can be continuously contracted to a point inside its own half-space without ever leaving it, yielding a link homotopy to a disjoint pair of points whose linking number is zero. We prove that width-3 feedforward networks with ReLU activations *preserve* linking numbers: invertible affine layers preserve $\text{link}$ as homeomorphisms, and monotonic activations preserve $\text{link}$ via straight-line homotopies. Since the Hopf link has $\text{link} = 1 \neq 0$, no such network can achieve linear separability, regardless of depth. Similarly, decision regions cannot be made disjoint and convex, so logit-based classifiers cannot achieve perfect accuracy.

How do modern architectures escape from such topological traps? The key is *folding*: coordinate-wise nonmonotonic transformations like the absolute value $|x|$. A ResNet block can synthesize $|x| = x + 2 \cdot \text{ReLU}(-x)$ using only ReLU and skip connections. Nonmonotonic activations (GELU, Swish) fold directly. Attention mechanisms create input-dependent convex combinations that locally approximate folding. These mechanisms break the homotopy argument underlying our impossibility theorems, enabling topological transformations that monotonic networks cannot perform.

**Our contributions.** We study the expressivity and learnability of neural network with low-dimensional topology:

1. **Topology–ML connection.** We prove that classification of linked data *requires* unlinking before a linear readout (Proposition 3.6).

2. **Topological expressivity.** We prove that width-$d$ feedforward networks with continuous coordinate-wise monotonic activations cannot separate linked manifolds in $\mathbb{R}^d$ (Sections 3–4). As a corollary, universal approximation with such activations requires width at least $d + 1$. We then show that skip connections, attention, nonmonotonic activations, and width expansion all provide ways to fold the data representation and thereby unlink linked manifolds (Section 5).

3. **Learnability beyond expressivity.** Experiments on Hopf links and higher-dimensional linked spheres show that limited topological expressivity imposes an accuracy ceiling under our training protocol. Further experiments on synthetic data and CIFAR-10 show that architectures with folding mechanisms gain a classification advantage, and this advantage is more pronounced at data points near the links (Section 6).

4. **Extrinsic TDA.** We present an algorithm for estimating linking between class manifolds from point-cloud samples, using spatial graphs, cycle bases, and Gauss integrals. Applied to CIFAR-10, the algorithm shows that class pairs with stronger linking tend to be harder for topologically less expressive architectures to classify (Section 6.6).

Table 1 offers a preview of our main expressivity results: which architectures can perform which topological transformations under width constraints. We will develop the theory in Sections 3–5, validate predictions experimentally in Section 6, and demonstrate real-world relevance via CIFAR-10 linking detection in Section 6.6.

*Table 1.* Topological expressivity of width-$d$ architectures, scored on the linking/folding transformations studied here. ✗ = cannot perform under our hypotheses; ✓ = can perform via the construction we give. **AH**: Ambient homeomorphism (flow-based models, Neural ODEs). **FM**: Feedforward monotonic (ReLU, sigmoid, tanh). **AE**: Autoencoder with width-$d$ bottleneck. **FN**: Feedforward nonmonotonic (GELU, Swish). **R**: ResNet. **T**: Pure transformer (two-token attention construction of Theorem 5.3).

| Topological Transformation | AH | FM | AE | FN | R | T |
|---|---|---|---|---|---|---|
| Deform shapes (no topological change) | ✓ | ✓ | ✓ | ✓ | ✓ | ✓ |
| Merge connected components | ✗ | ✓ | ✓ | ✓ | ✓ | ✓ |
| Fill up holes | ✗ | ✓ | ✓ | ✓ | ✓ | ✓ |
| Unlink 2-component link | ✗ | ✗ | ✗ | ✓ | ✓ | ✓ |

## 2. Related Work

Topological data analysis extracts intrinsic shape features (holes, connected components) via persistent homology (Carlsson, 2009; Edelsbrunner & Harer, 2010; Cang & Wei, 2017). Naitzat et al. (2020) study how deep networks change Betti numbers of data manifolds; we study how narrow networks *preserve* linking numbers, an extrinsic invariant depending on how manifolds are embedded in an ambient space, not an intrinsic invariant of a manifold itself like the Betti numbers. This extrinsic perspective connects to early intuitions about neural networks "untangling" data (Olah, 2014). Recent works have identified other topological limitations: higher-order message-passing cannot compute homology (Eitan et al., 2025), E(3)-invariant GNNs cannot distinguish enantiomers (Dumitrescu et al., 2025), and topological obstructions constrain generative models (Esmaeili et al., 2023). These concern what networks can *detect*; we study what they can *transform*, a question also relevant to world models that must represent geometric structure (LeCun, 2022). Width bounds for universal approximation (Hanin & Sellke, 2017; Johnson, 2019; Rochau et al., 2024) show that width-$d$ networks cannot approximate all functions on $\mathbb{R}^d$; we on the other hand show that arbitrarily deep width-$d$ networks cannot separate linked data in $\mathbb{R}^d$. Neural networks for molecular dynamics lift $\mathbb{R}^3$ coordinates to 64- to 384-dimensional features (Schütt et al., 2017; Satorras et al., 2021; Jumper et al., 2021); our theory offers a topological rationale for this practice.

## 3. The Hopf Link: An Intuitive Example

Consider two interlinked circles in 3D space forming the Hopf link (for topological background, see Rolfsen, 1976; Hatcher, 2002).

> Question: *Can a width-3 ReLU network classify points on these circles into separate classes?*

Here a width-$d$ feedforward network is $F = A_L \circ \sigma \circ \cdots \circ \sigma \circ A_1$ with affine $A_i(x) = W_i x + b_i$, $W_i \in \mathbb{R}^{d \times d}$, and coordi-

natewise activation $\sigma$, e.g., ReLU. Our results apply to any architecture whose intermediate representations have dimensions $\leq d$ after flattening any spatial axes and whose nonlinearities are coordinatewise. This includes CNNs, since convolutional layers are affine maps and the flattened representation size $C \cdot H \cdot W$ is the operative width, as well as bottleneck autoencoders with width-$d$ bottlenecks.

**Example 3.1** (The Hopf link). The *Hopf link* consists of two interlinked circles in $\mathbb{R}^3$ given parametrically by

$$X(t) = (\cos(t), \sin(t), 0),$$
$$Y(s) = (1 + \cos(s), 0, \sin(s)),$$

for $t, s \in [0, 2\pi]$. These two circles are topologically linked with $\text{link}(X, Y) = 1$, meaning they cannot be separated without cutting one of the curves.

Classifying samples from the two Hopf link circles exposes a topological constraint on narrow neural networks; we develop the argument step by step, leaving technical details to Appendix C.

### 3.1. Linking Numbers

The core of our argument is a topological invariant that quantifies how two closed curves are intertwined in 3D space.

**Definition 3.2** (Linking number). Let $X, Y \subset \mathbb{R}^3$ be two disjoint, oriented, simple closed curves. The *linking number* $\text{link}(X, Y)$ is defined by the Gauss integral:

$$\text{link}(X, Y) = \frac{1}{4\pi} \oint_X \oint_Y \frac{(x - y) \cdot (dx \times dy)}{|x - y|^3}.$$

This measures the degree of the Gauss map $G : X \times Y \to \mathbb{S}^2$ given by $G(x, y) = (x - y)/\|x - y\|$. The linking number is always an integer. For the Hopf link, $\text{link}(X, Y) = \pm 1$.

There is an equivalent combinatorial formula: For any regular projection $\pi : \mathbb{R}^3 \to \mathbb{R}^2$,

$$\text{link}(X, Y) = \frac{1}{2} \sum_{p \in \pi(X) \cap \pi(Y)} \epsilon_p$$

where $\epsilon_p = \pm 1$ is the sign of each crossing.

### 3.2. Network Operations and Linking Number Preservation

We analyze each layer type in a width-3 ReLU network through the following key lemmas.

**Definition 3.3** (Homotopy). A *homotopy* between two continuous maps $f, g : X \to Z$ is a continuous function $H : X \times [0, 1] \to Z$ such that $H(x, 0) = f(x)$ and $H(x, 1) = g(x)$ for all $x \in X$. Intuitively, $H$ provides a continuous "interpolation" from $f$ to $g$, deforming $f(X)$ to $g(X)$ within $Z$. The *straight-line homotopy*

_Figure 2._ Rank-deficient transformations force intersection.

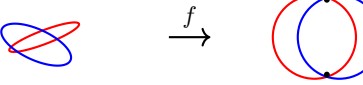

$\mathbb{R}^3$: link $\neq 0$      $\mathbb{R}^2$: intersection

$H_t(x) = (1-t)f(x) + tg(x)$ is the simplest example when $Z = \mathbb{R}^n$.

**Lemma 3.4** (Homotopy invariance). _If $X'$ is homotopic to $X$ via a homotopy avoiding $Y$, then_

$$\text{link}(X', Y) = \text{link}(X, Y).$$

_Proof sketch._ The function $t \mapsto \text{link}(X_t, Y)$ is continuous and integer-valued on $[0, 1]$, hence constant.    $\square$

**Lemma 3.5** (Rank-deficient intersection). _Let $X, Y \subset \mathbb{R}^3$ be disjoint curves with $\text{link}(X, Y) \neq 0$. For any rank-deficient affine $f : \mathbb{R}^3 \to \mathbb{R}^3$, we have $f(X) \cap f(Y) \neq \varnothing$._

_Proof sketch._ Reduce via SVD to projection. If projected curves were disjoint, the combinatorial formula gives $\text{link} = 0$, contradiction.    $\square$

### 3.3. Classification Requires Unlinking

Linear separability is the goal of representation learning for classification: the final layer (classifier head) applies a linear transformation to features, so perfect classification requires that class representations become linearly separable. The task imposes a strict topological constraint:

**Proposition 3.6** (Linear readouts require unlinking). _Let $X, Y \subset \mathbb{R}^3$ be disjoint oriented closed curves with $\text{link}(X, Y) \neq 0$. If a continuous feature map $F : \mathbb{R}^3 \to \mathbb{R}^3$ makes $F(X)$ and $F(Y)$ linearly separable, then $\text{link}(F(X), F(Y)) = 0$; the feature map must unlink the classes._

_Proof._ A separating hyperplane $H$ puts $F(X)$ in one open half-space and $F(Y)$ in the other. Each half-space is convex, so straight-line contractions of $F(X)$ and $F(Y)$ to points $p$ and $q$ inside their respective half-spaces stay disjoint. This gives a link homotopy to two point components. At the endpoint, the Gauss map is constant, hence has degree zero; by link-homotopy invariance, $\text{link}(F(X), F(Y)) = 0$.    $\square$

This exposes the conflict: to classify the Hopf link perfectly, a network must change the linking number from $\pm 1$ to 0.

### 3.4. Separation Impossibility for the Hopf Link

**Theorem 3.7** (Link separation impossibility). _Let $X, Y \subset \mathbb{R}^3$ be disjoint simple closed curves with $\text{link}(X, Y) \neq 0$. Let $F : \mathbb{R}^3 \to \mathbb{R}^3$ be any width-3 feedforward network with affine transformations and ReLU activations. Then $F(X)$ and $F(Y)$ are not linearly separable, and perfect classification is impossible._

_Proof sketch._ Assume for contradiction that linear separability is achieved. Proposition 3.6 gives $\text{link}(F(X), F(Y)) = 0$; moreover, any intermediate collision between the two class images would propagate through later layers and prevent final linear separation, so the class images stay disjoint throughout the network. We analyze each layer:

**Invertible affine layers:** Homeomorphisms of $\mathbb{R}^3$ preserve linking numbers (up to sign).

**ReLU activations:** We apply homotopies _one coordinate at a time_. For coordinate $i$, define $H_t^{(i)}$ that interpolates $x_i \to \text{ReLU}(x_i)$ while other coordinates stay fixed. Suppose paths collide: $H_t^{(i)}(x)_i = H_t^{(i)}(y)_i$ for some $t$. If $x_i \geq 0$, the path is constant at $x_i$; if $x_i < 0$, it traverses $[x_i, 0]$. In all cases, collision implies $\text{ReLU}(x_i) = \text{ReLU}(y_i)$. Since $\sigma(X) \cap \sigma(Y) = \varnothing$ by assumption, no collision occurs. Concatenating $H^{(1)}, H^{(2)}, H^{(3)}$ gives $\text{link}(\sigma(X), \sigma(Y)) = \text{link}(X, Y)$.

**Rank-deficient layers:** By Lemma 3.5, these create intersections when $\text{link} \neq 0$.

Since ReLU preserves link and invertible affines preserve it up to sign, changing it from non-zero to zero requires rank-deficient layers. But these force intersections.    $\square$

This example demonstrates the core mechanism: linking numbers create barriers that narrow monotonic networks cannot overcome, regardless of depth. The next section generalizes to higher-dimensional linked manifolds and arbitrary coordinate-wise monotonic activations.

## 4. Higher Dimension, More Activations

The Hopf link example gives the core mechanism in a visual setting. In this section we switch notation from the curves $X, Y \subset \mathbb{R}^3$ of Section 3 to manifolds $M^m, N^n \subset \mathbb{R}^d$, and generalize from ReLU to arbitrary coordinate-wise monotonic activations.

### 4.1. Higher-Dimensional Linking Numbers

The concept of linking generalizes from curves to higher-dimensional manifolds. For two closed, oriented, disjoint manifolds $M^m$ and $N^n$ in ambient space $\mathbb{R}^d$ where $d =$

$m + n + 1$, the linking number is well-defined via degree.

In the layerwise arguments below, $F$ denotes the composition of the first $j$ network layers, for some $j$, applied to both manifolds. The images $F(M)$ and $F(N)$ may no longer be manifolds; whenever they remain disjoint, $\text{link}(F(M), F(N))$ means the linking number of the maps $F \circ \iota_M$ and $F \circ \iota_N$ parametrizing the images $F(M)$ and $F(N)$.

**Definition 4.1** (Higher-dimensional linking number). For two disjoint, closed, oriented manifolds $M^m$ and $N^n$ in $\mathbb{R}^d$ where $d = m + n + 1$, the linking number $\text{link}(M, N)$ is defined via the Gauss map $G : M \times N \to \mathbb{S}^{d-1}$ given by $G(x, y) = (x - y)/|x - y|$:

$$\text{link}(M, N) = \deg(G)$$

where $\deg(G)$ is the topological degree.

**Example 4.2** (Higher Hopf links: $\mathbb{S}^n \sqcup \mathbb{S}^n$ in $\mathbb{R}^{2n+1}$). Two $n$-spheres can be linked in $\mathbb{R}^{2n+1}$ with $\text{link} = \pm 1$. For $n = 1$, this recovers the classical Hopf link ($\mathbb{S}^1 \sqcup \mathbb{S}^1 \subset \mathbb{R}^3$). For $n = 2$, we obtain linked 2-spheres $\mathbb{S}^2 \sqcup \mathbb{S}^2 \subset \mathbb{R}^5$, which we use in experiments (Section 6.5). The construction embeds each sphere into complementary coordinate subspaces via stereographic projection.

The key lemmas from Section 3 generalize:

**Lemma 4.3** (Higher-dimensional homotopy invariance). *Let $M^m, N^n \subset \mathbb{R}^d$ (with $d = m + n + 1$) be disjoint manifolds. If $M'$ is homotopic to $M$ via a homotopy avoiding $N$, then $\text{link}(M', N) = \text{link}(M, N)$.*

**Lemma 4.4** (Higher-dimensional linear separability). *If $M^m$ and $N^n$ in $\mathbb{R}^d$ (with $d = m + n + 1$) are linearly separable, then $\text{link}(M, N) = 0$.*

**Theorem 4.5** (Higher-dimensional separation impossibility). *Let $M^m$ and $N^n$ be disjoint, closed, oriented manifolds in $\mathbb{R}^d$ (with $d = m + n + 1$) such that $\text{link}(M, N) \neq 0$. Then no width-$d$ feedforward network with affine transformations and coordinate-wise monotonic activations can linearly separate $M$ and $N$, and perfect classification is impossible.*

*Proof sketch.* The argument follows Theorem 3.7: invertible affines preserve $\text{link}$ up to sign as homeomorphisms; monotonic activations preserve $\text{link}$ via the same straight-line homotopy argument (which works in any dimension); rank-deficient transformations force intersections when $\text{link} \neq 0$. $\square$

### 4.2. General Monotonic Activations

The Hopf link analysis used ReLU, but the impossibility extends to any *coordinate-wise monotonic* activation $\sigma(x) = (\sigma_1(x_1), \ldots, \sigma_n(x_n))$ where each $\sigma_i$ is continuous and monotonic, e.g., sigmoid, tanh, Leaky ReLU, ELU. This

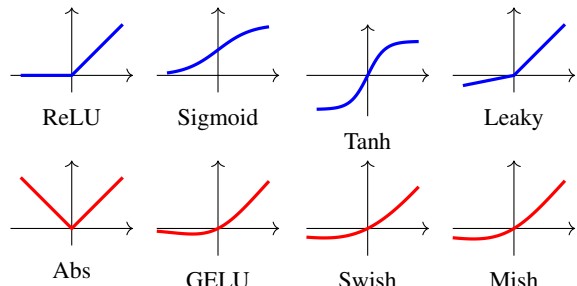

*Figure 3.* Monotonic (top) vs nonmonotonic (bottom) activations.

shows the obstruction is fundamental to monotonicity, not an artifact of ReLU's piecewise-linear form.

The key to generalizing beyond ReLU is the concept of *link homotopy*: a simultaneous continuous deformation of multiple components that keeps them disjoint throughout. Unlike single-component homotopy, where one set moves while others stay fixed, link homotopy moves all components together via the same formula $H_t(x) = (1 - t)x + t\sigma(x)$. This simultaneity is essential: if only one component moves, it can collide with stationary components even for nondecreasing activations.

**Lemma 4.6** (Monotonic activation preservation). *Let $\sigma : \mathbb{R}^d \to \mathbb{R}^d$ be coordinate-wise monotonic with $r$ nonincreasing coordinates. If $M, N \subset \mathbb{R}^d$ are disjoint compact manifolds such that $\sigma(M)$ and $\sigma(N)$ remain disjoint, then:*

$$\text{link}(\sigma(M), \sigma(N)) = (-1)^r \text{link}(M, N)$$

*Proof sketch.* The straight-line homotopy $H_t(x) = (1 - t)x + t\sigma(x)$ applied simultaneously to both $M$ and $N$ defines a link homotopy when $\sigma$ is nondecreasing: if $x_i < y_i$, then monotonicity gives $\sigma_i(x_i) \leq \sigma_i(y_i)$, so $H_t(x)_i < H_t(y)_i$ for $t \in [0, 1)$, and the assumed disjointness $\sigma(M) \cap \sigma(N) = \varnothing$ at $t = 1$ rules out a collision at the endpoint. For nonincreasing coordinates, we decompose $\sigma = R \circ \sigma'$ where $\sigma'$ is nondecreasing and $R$ is the output reflection negating those coordinates; reflections contribute the sign factor $(-1)^r$. $\square$

**Theorem 4.7** (General impossibility theorem). *Let $M^m, N^n \subset \mathbb{R}^d$ be disjoint closed oriented submanifolds with complementary dimension $m + n + 1 = d$ and $\text{link}(M, N) \neq 0$. Let $F : \mathbb{R}^d \to \mathbb{R}^d$ be any width-$d$ feedforward network with affine transformations and coordinate-wise monotonic activations. Then $F(M)$ and $F(N)$ are not linearly separable, and perfect classification is impossible.*

*Proof sketch.* By Lemma 4.6, monotonic activations preserve $\text{link}$ up to sign via link homotopy. The proof structure of Theorem 3.7 carries through: invertible affines preserve $\text{link}$ up to sign, monotonic activations preserve $\text{link}$ up to sign, and since $\text{link} \neq 0$ implies $\pm \text{link} \neq 0$, rank-deficient

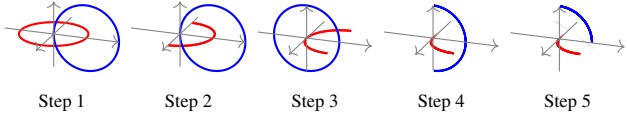

Step 1     Step 2     Step 3     Step 4     Step 5

*Figure 4.* Hopf link unlinking via $|x|$ activations (full resolution: Figure 9).

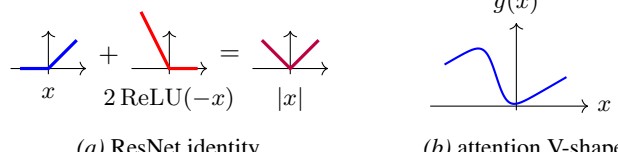

*(a)* ResNet identity      *(b)* attention V-shape

*Figure 5.* Mechanisms for expressing non-monotonicity in coordinates with monotonic activation functions.

layers are required to achieve $\mathrm{link} = 0$. But rank-deficient layers force intersections. $\square$

The result applies equally to smooth activations (sigmoid, tanh) and piecewise-linear activations (ReLU): the crucial property is monotonicity, not differentiability. Thus the topological barrier is intrinsic to the coordinate-wise monotonic structure.

# 5. Breaking Topological Constraints: Architectural Mechanisms

The impossibility results in Sections 3–4 demonstrate fundamental limitations of narrow networks with coordinate-wise monotonic activations. These constraints can be overcome through architectural approaches while preserving narrow width. We explore four complementary mechanisms, with supporting constructions in Appendices D and F.

## 5.1. Non-Monotonic Activation Functions

The first approach breaks the monotonicity constraint underlying our impossibility theorems. While ReLU and other monotonic activations preserve topological rigidity, non-monotonic functions create geometric flexibility needed to overcome linking obstructions.

Important nonmonotonic activations (Figure 3, bottom) include GELU (Hendrycks & Gimpel, 2016) ($x \cdot \Phi(x)$, standard in Transformers), Swish/SiLU (Ramachandran et al., 2017; Elfwing et al., 2018) ($x \cdot \sigma(x)$), and Mish (Misra, 2020) ($x \cdot \tanh(\mathrm{softplus}(x))$). The impossibility results critically depend on Lemma 3.4: monotonic activations preserve linking numbers via straight-line homotopies that avoid intersections. This fails for nonmonotonic functions.

**Example 5.1** (Nonmonotonic unlinking). Coordinate-wise $|x|$ can unlink the Hopf link through "folding" operations that collapse signed coordinates into the positive octant. By translating data appropriately, each folding layer affects only one coordinate at a time: $(x_1, x_2, x_3) \mapsto (|x_1|, x_2, x_3) \mapsto (|x_1|, |x_2|, x_3) \mapsto \cdots$ Figure 4 illustrates.

The same recipe extends to any activation $\sigma$ that has a strict local extremum on an open interval $I \subset \mathbb{R}$, e.g., GELU near $-0.5$, Swish near $-1.3$, Mish near a similar point: for compact data, affine layers rescale the relevant coordinate into $I$, and the nonmonotonicity of $\sigma|_I$ breaks the same

homotopy-disjointness argument that fails for $|\cdot|$.

## 5.2. Skip Connections and Residual Networks

Residual networks (He et al., 2016) maintain monotonic activations but break the purely feedforward constraint through skip connections. The key insight is that ResNet can express nonmonotonic functions:

**Theorem 5.2** (ResNet topological expressivity). *Width-$n$ ResNet architectures using only ReLU activations can perform the same topological transformations as networks with nonmonotonic activations, including unlinking.*

*Proof sketch.* ResNet blocks $\mathcal{F}(x) = x + \mathcal{G}(x)$ can express absolute value via the identity:

$$|x| = x + 2\,\mathrm{ReLU}(-x) \tag{1}$$

Setting $\mathcal{G}(x) = 2\,\mathrm{ReLU}(-x)$ yields $\mathcal{F}(x) = |x|$. This enables the same coordinate-wise folding operations as nonmonotonic activations. $\square$

While ResNets can be viewed as discretizations of Neural ODEs (Chen et al., 2018), the discrete formulation is strictly more expressive for topological transformations. Neural ODEs generate continuous flows (diffeomorphisms) that must preserve topological invariants. Discrete ResNet blocks can implement folding maps that change these invariants; the $|x| = x + 2\,\mathrm{ReLU}(-x)$ identity relies crucially on discrete, non-infinitesimal residuals.

## 5.3. Attention Mechanisms

The Transformer architecture (Vaswani et al., 2017) introduces self-attention mechanisms enabling global information routing. We study the topological expressivity of attention by considering *pure transformers*.

**Theorem 5.3** (Two-token attention as a coordinate fold). *For each input coordinate $x_i$ there is a single-head two-token attention layer (with distinct positional encodings $(p_1, p_2)$ and scalar query/key/value/output weights) whose output $g(x_i)$ has a strict local minimum at $x_i = 0$ and is monotonically decreasing on a left interval and monotonically increasing on a right interval, giving a smoothed V-shape that approximates $|x_i|$ near the origin. Applying*

*this construction coordinate-wise yields a pure-transformer realization of the coordinate-wise fold used in our unlinking construction, after an affine pre-shift that rescales the data into the V-shape's effective region.*

*Proof sketch.* Use a two-token input $(x_i, x_i)$ (or a copy-token preamble): with distinct positional encodings the second-position attention weight reduces to a sigmoid in $q_2(k_1 - k_2)$, and an explicit choice of scalar weights produces the V-shape (Figure 5b). The construction is a per-coordinate *local* surrogate for $|\cdot|$, not a global approximation; it suffices for the topological transformation because the unlinking construction depends only on the existence of a coordinate-wise nonmonotonic fold on the data domain, not on exact equality with $|\cdot|$. □

### 5.4. Width Threshold and Design Implications

**Width $\geq d + 1$ eliminates the classification obstruction.** Our impossibility results are tight: width-$(d + 1)$ ReLU networks achieve universal approximation on compact subsets of $\mathbb{R}^d$ (Hanin & Sellke, 2017), and subsequent work sharpens minimum-width thresholds for ReLU, leaky-ReLU, and compact-domain settings (Park et al., 2021; Cai, 2023; Li et al., 2023; Kim et al., 2024). This suffices to map any disjoint configuration of class manifolds into disjoint scalar class labels, i.e., to achieve linear separability; it does not require "unlinking" as an ambient-topology operation, only producing the right value of a continuous label function.

**Design guidance.** For topologically complex data, width should be read locally as well as globally: a wide model may contain concrete narrow maps, such as the $d_{\text{model}} \to d_k, d_v$ projections in a single attention head, a latent bottleneck, or the reduced channel inside a bottleneck residual block. In such settings: (1) expand MLP width slightly beyond input dimension before narrowing; (2) prefer nonmonotonic activations (GELU, SiLU, Mish) in bottleneck layers; (3) use skip connections; or (4) leverage attention. These mechanisms change topology directly rather than relying solely on dimension expansion.

## 6. Experiments

We validate our theoretical results with experiments on the Hopf link classification task. These experiments verify both the impossibility results for monotonic activations and the escape mechanisms described in the previous section, with experimental and detection details in Appendices G–I.

### 6.1. Experimental Setup

**Thickened Hopf link.** We construct volumetric data by sampling points from two thickened interlinked tori. Each curve is thickened by adding small perturbations normal to

*Table 2.* Held-out accuracy (%) on Hopf link classification across depths, ReLU vs GELU (30 seeds per cell). Rows report mean, std, and maximum over seeds.

| Model | Stat | 3 | 5 | 8 | 12 | 16 | 20 |
|-------|------|------|------|------|------|------|------|
| | mean | 84.3 | 77.1 | 63.1 | 57.7 | 53.5 | 50.3 |
| ReLU | std | 9.1 | 18.7 | 18.1 | 15.1 | 9.9 | 1.8 |
| | max | 92.8 | 92.5 | 92.6 | 91.6 | 91.4 | 54.4 |
| | mean | 89.3 | 90.0 | 91.1 | 91.2 | 72.8 | 52.8 |
| GELU | std | 2.6 | 2.8 | 3.1 | 2.0 | 19.7 | 9.8 |
| | max | 92.9 | 100.0 | 100.0 | 100.0 | 92.6 | 90.1 |

*Table 3.* Held-out accuracy (%) on Hopf link classification across depths, plain ReLU vs ResNet (30 seeds per cell). Rows report mean, std, and maximum over seeds.

| Model | Stat | 3 | 4 | 5 | 6 | 7 | 8 |
|-------|------|------|------|------|------|------|------|
| | mean | 83.9 | 76.5 | 74.3 | 69.9 | 64.8 | 66.2 |
| Plain | std | 10.1 | 15.2 | 18.1 | 19.9 | 18.7 | 19.0 |
| | max | 92.0 | 92.3 | 91.2 | 91.9 | 91.5 | 91.5 |
| | mean | 97.5 | 97.3 | 98.5 | 96.6 | 97.5 | 97.5 |
| ResNet | std | 4.4 | 3.6 | 2.4 | 3.8 | 3.1 | 3.0 |
| | max | 100.0 | 100.0 | 100.0 | 100.0 | 100.0 | 100.0 |

the curve surface. Dataset: 6000 points (3000 per class), split 80/20 train/validation.

**Training protocol.** Adam/AdamW with learning rate $10^{-3}$, up to 800 epochs with early stopping (patience 100–200). Cross-entropy loss.

### 6.2. ReLU vs GELU on Hopf Link

We compare monotonic (ReLU) versus nonmonotonic (GELU) activations across 12 depths (3–20 layers) with 30 random seeds each (720 total experiments).

Table 2 confirms the theory: ReLU stays at or below the topological ceiling and degrades with depth as optimization difficulty compounds the expressivity barrier (Theorem 3.7); GELU consistently achieves ∼90% at moderate depths, breaking the barrier. At extreme depths ($\geq 16$) both activations suffer optimization instability (large stds). Across 30 seeds, GELU's best run reaches 100.0% at depths 5–12 while ReLU never exceeds 92.8%, matching the prediction.

### 6.3. ResNet vs Plain ReLU

We compare plain ReLU feedforward networks against ReLU ResNets, both at width 3.

Across 30 seeds, ResNet reaches a **100% best run at every depth from 3 to 8**, and mean accuracy stays at 96.6–98.5% across all depths. Plain ReLU never exceeds 92.3% and has much lower means with high seed variance, confirming both

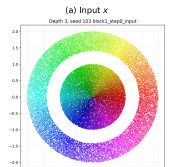 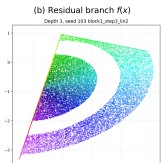 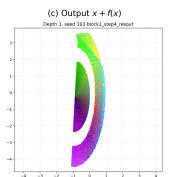

*Figure 6.* ResNet skip connection implementing $|x| = x + 2\operatorname{ReLU}(-x)$ on linked disk-annulus (pt-$\mathbb{S}^1$ link). (a) Input $x$. (b) Residual branch $f(x) \approx 2\operatorname{ReLU}(-x)$. (c) Output $x + f(x)$: folding separates components.

*Table 4.* Best test accuracy (%) across 100 seeds for total linking number $k$, i.e., $k$ disjoint copies of linked $\mathbb{S}^2 \sqcup \mathbb{S}^2$, in $\mathbb{R}^5$. Width-5, depth-5 networks; "best" reports the expressivity ceiling per cell.

| Model | $k=1$ | $k=2$ | $k=5$ | $k=10$ | $k=20$ | $k=50$ |
|---|---|---|---|---|---|---|
| ReLU | 98.6 | 98.0 | 93.5 | 87.9 | 84.5 | 80.2 |
| ReLU+Skip | **100.0** | **98.7** | **94.1** | 85.4 | 83.6 | 80.9 |
| GELU | 98.8 | 98.4 | 92.5 | 91.1 | 88.6 | **84.8** |
| Swish | 97.0 | 98.2 | 92.0 | **91.6** | **88.9** | 84.3 |

an expressivity barrier and depth-dependent optimization difficulty. Skip connections provide a mechanism to overcome topological barriers even with monotonic activations, as predicted by Theorem 5.2.

### 6.4. Mechanistic Interpretability: How ResNet Unlinks a Point/Circle Pair in $\mathbb{R}^2$

The disk-annulus separation task is the ambient-dimension $d = 2$ instance of our linking framework, given by Theorem 4.7 with $m = 0$, $n = 1$, and $d = 2$: a point inside the annulus and the annulus' inner boundary circle form a 0-manifold/1-manifold link with $|\operatorname{link}| = 1$, equivalently winding number $\pm 1$. By our theory, width-2 monotonic feedforward networks cannot linearly separate them, but a width-2 ResNet can.

Figure 6 shows the layer-by-layer transformation. ResNet implements "folding" operations via $|x| = x + 2\operatorname{ReLU}(-x)$ (Equation 1), enabling nonmonotonic transformations that separate the topologically linked components.

### 6.5. Higher-Dimensional Linking: $\mathbb{S}^n \sqcup \mathbb{S}^n$ in $\mathbb{R}^{2n+1}$

We extend experiments to higher dimensions using linked hyperspheres: two $n$-spheres embedded in $\mathbb{R}^{2n+1}$ with linking number $\pm 1$. For $n = 2$, this gives $\mathbb{S}^2 \sqcup \mathbb{S}^2 \subset \mathbb{R}^5$ classified by width-5 networks. We quantify topological difficulty via linking number: a single link has only one local entanglement region, so networks can achieve high accuracy by being wrong only in that small patch; placing $k$ disjoint copies forces the network to handle larger entanglement regions (Table 4).

At small $k$, monotonic architectures with skip connections

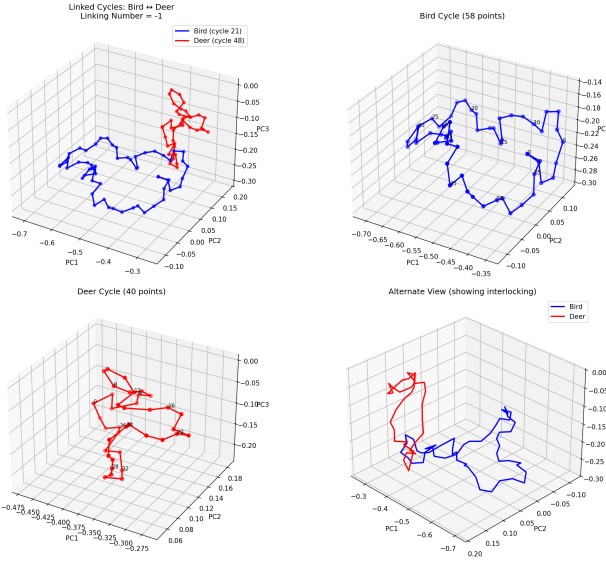

*Figure 7.* Linked cycles in CIFAR-10: bird (blue) and deer (red) with $\operatorname{link} = -1$ at $\varepsilon = 0.034$.

reach the expressivity ceiling (ReLU+Skip attains 100% at $k = 1$, 98.7% at $k = 2$). As $k$ increases, the best nonmonotonic architectures (GELU/Swish) hold a 3–4pp accuracy advantage over the best monotonic architecture at $k \geq 10$, consistent with nonmonotonic activations being able to resolve each local entanglement separately. The effect at $k = 1$ is weaker than for $n = 1$ (Table 2) due to the linking region's smaller volume fraction in higher dimensions.

### 6.6. Linking in Real Data: CIFAR-10

As a suggestive proof-of-concept, we investigate whether topological linking detectable by our algorithm corresponds to classification difficulty in real image data. We emphasize that the evidence below is correlational rather than causal: classification on real images involves many confounding factors beyond ambient topology, and the construction relies on 3D PCA projection rather than the native data manifold. Using the linking detection algorithm, we analyze CIFAR-10 class manifolds projected to 3D via PCA.

**Link detection.** With $20\times$ data augmentation (1.05M samples) and $k$-NN graph construction ($k = 15$, mutual edges), we detect linked cycles between bird and deer classes at $\varepsilon = 0.034$ (0.22% of the 3D bounding box diagonal). The witness cycles have linking number $\operatorname{link} = -1$ with Gauss integral $-1.004$ (Figure 7). This binary search only locates the onset scale; the all-pair consistency study uses fixed thresholds, and detected graph-cycle witnesses persist as $\varepsilon$ increases because the filtered $k$-NN edge set only grows. This is a reproducible linking signal in PCA-3D, not a claim about linking of the full 3072D pixel manifold.

**10-class linking consistency.** Running detection 11 times

*Table 5.* Binary classification accuracy at L8 no-skip: linked pair (deer-dog) vs unlinked control (frog-ship). All 3 nonmonotonic activations outperform all 4 monotonic activations on the linked pair; no such pattern on the unlinked pair.

| Activation | Linked (deer-dog) | | Unlinked (frog-ship) | |
|---|---|---|---|---|
| | Type | Acc (%) | Type | Acc (%) |
| GELU | NONM | **90.8** | NONM | 97.9 |
| Mish | NONM | **90.8** | NONM | 97.7 |
| Swish | NONM | 90.1 | NONM | 97.8 |
| ReLU | MONO | 89.6 | MONO | 97.8 |
| LeakyReLU | MONO | 89.6 | MONO | 97.7 |
| ELU | MONO | 89.1 | MONO | 97.2 |
| SELU | MONO | 87.4 | MONO | 97.6 |
| **Gap** | +1.2% | | +0.1% | |

across all 45 class pairs yields a linking-consistency summary. Of 45 pairs, 27 are strongly linked (>70% consistency), 8 are weak or unlinked (<30%), and 10 are ambiguous. High examples include deer–dog (91%), bird–cat (91%), cat–dog (82%), and automobile–truck (91%); low controls include frog–ship (18%), airplane–horse (18%), and cat–ship (27%).

**Classification experiments.** We train width-bounded CNNs (all intermediate layers $\leq$ 3072D) on both binary and 10-class classification tasks, comparing monotonic (ReLU, ELU, SELU, LeakyReLU) versus nonmonotonic (GELU, Swish, Mish) activations.

At L8 no-skip (Table 5), all nonmonotonic activations outperform all monotonic activations on the linked deer-dog pair (+1.2% gap), while no such pattern exists on the unlinked frog-ship pair (control). With skip connections, the effect disappears: monotonic networks match nonmonotonic performance, as predicted by Theorem 5.2. As a localization diagnostic, we retrained bird–deer L8 no-skip ReLU/GELU classifiers, detected one PCA-3D link witness, and stratified the unaugmented test set by distance to that witness. The activation gap is largest near the witness and decays with distance (Table 6); this is diagnostic rather than causal evidence, since distance is measured in PCA-3D and can mix topology with local geometry. PCA-3D linking consistency also correlates with within-CIFAR class-pair confusion ($r \approx 0.48$, $p < 0.001$), outperforming pixel-space distance metrics.

*Table 6.* Bird–deer test accuracy by distance to a detected PCA-3D link witness.

| Subset | $n$ | ReLU (%) | GELU (%) | Gap (pp) |
|---|---|---|---|---|
| $< 10\varepsilon$ | 31 | 90.6 | 97.1 | +6.5 |
| $< 20\varepsilon$ | 243 | 89.4 | 92.3 | +2.9 |
| $< 50\varepsilon$ | 1191 | 89.3 | 91.4 | +2.1 |
| All | 2000 | 89.9 | 91.6 | +1.6 |
| $> 50\varepsilon$ | 809 | 90.6 | 91.4 | +0.8 |

**Expressivity vs. optimization.** Our theoretical results concern *expressivity*, meaning what functions network architectures can represent, not *optimization*, meaning what functions gradient descent actually finds. Even when two architectures have identical expressivity ceilings, they may exhibit different training dynamics: convergence speed, sensitivity to learning rate, propensity to find particular local minima, etc. The observed CIFAR-10 gaps thus reflect a combination of (i) the topological expressivity barrier predicted by our theory, and (ii) optimization dynamics that our theory does not address.

## 7. Conclusion

This work develops a new lens for neural network expressivity analysis: from approximation rates to geometric transformation requirements. Rather than asking whether a network can approximate a target function, we ask what geometric operations on embedded data it can perform. For the pairwise linking and folding phenomena studied here, this perspective extends to mechanistic understanding: ResNet skip connections, attention mechanisms, and nonmonotonic activations all break the monotonicity required by our impossibility argument via a "folding" construction.

Our framework focuses on *ambient-sensitive* (extrinsic) topological invariants, fundamentally distinguishing it from traditional topological data analysis which studies intrinsic properties. The linking number of two manifolds depends on *how* they are embedded in $\mathbb{R}^n$, not merely on the manifolds themselves. This extrinsic perspective captures the algorithmic reality: neural networks must transform data within the ambient space, making embedding geometry as important as intrinsic data structure.

We expect this work to open broader applications of low-dimensional topology and geometric topology to machine learning. The invariant we study, the pairwise linking number, is an initial example. A companion paper treats richer invariants from link theory and knot theory, such as Milnor's $\bar{\mu}$-invariants and knot types, in detail (Ren & Lim, 2026). Similarly, the task correspondence we establish between unlinking and classification likely represents an initial example of a broader pattern yet to be fully explored. Understanding this interplay between data geometry and algorithmic capability offers new directions for both theoretical foundations and principled architecture design.

## Impact Statement

This paper presents theoretical and empirical work on neural network expressivity. We identify fundamental limitations of certain architectures, which may inform more principled architecture design. We also propose a link detection algorithm that characterizes topological complexity in datasets,

which may inform dataset difficulty analysis.

## Acknowledgments

JR and LH are partially supported by a Vannevar Bush Faculty Fellowship ONR N000142312863.

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

# A. Network Architecture

This appendix collects the network-architecture definitions used in the main text (§A.1) and the classifier-head/linear-separability equivalence (§A.2).

## A.1. Network Architectures

**Definition A.1** (Width-$d$ feedforward network). A *width-$d$ feedforward network* with activation $\sigma$ is $F = A_L \circ \sigma \circ A_{L-1} \circ \sigma \circ \cdots \circ \sigma \circ A_1$, where each $A_i(x) = W_i x + b_i$ is affine with $W_i \in \mathbb{R}^{d \times d}$, $b_i \in \mathbb{R}^d$, and $\sigma$ is applied coordinate-wise. All intermediate representations lie in $\mathbb{R}^d$.

**Definition A.2** (Width-$d$ ResNet). A *width-$d$ ResNet* is $F = B_L \circ \cdots \circ B_1$ where each block $B_i(x) = x + R_i(x)$ and $R_i = A_{i,2} \circ \sigma \circ A_{i,1}$ is a width-$d$ feedforward sublayer with monotonic $\sigma$ (typically ReLU).

**Definition A.3** (Pure transformer). A *pure transformer* is a transformer with residual connections and layer normalization removed: each block applies self-attention $\text{Attention}(X) = \text{softmax}\left(\frac{X W_Q (X W_K)^T}{\sqrt{d_k}}\right) X W_V$, then an affine layer, then a coordinate-wise activation. The width constraint requires all intermediate dimensions $\leq n$.

**Definition A.4** (Autoencoder with bottleneck $d$). An *autoencoder with bottleneck $d$* is $F = D \circ E$ where the encoder $E : \mathbb{R}^n \to \mathbb{R}^d$ projects via an initial affine $A_1 : \mathbb{R}^n \to \mathbb{R}^d$ followed by width-$d$ monotonic-feedforward layers, and the decoder $D : \mathbb{R}^d \to \mathbb{R}^n$ symmetrically expands via width-$d$ layers and a final $A_L : \mathbb{R}^d \to \mathbb{R}^n$.

**Corollary A.5** (Autoencoder topological impossibility). *Let $X, Y \subset \mathbb{R}^n$ be disjoint compact manifolds contained in a $d$-dimensional affine subspace $H \subset \mathbb{R}^n$ with $\text{link}(X, Y) \neq 0$. No autoencoder with bottleneck $d$ and coordinate-wise monotonic activations can transform $(X, Y)$ into linearly separable images while preserving disjointness.*

*Proof.* After an appropriate rotation, assume $H = \mathbb{R}^d \times \{0\}^{n-d}$. Let $G : H \cong \mathbb{R}^d \to \mathbb{R}^d$ denote the pre-final-decoder feature map, i.e., the composition of the encoder restricted to $H$ with all width-$d$ decoder layers *except* the final affine $A_L : \mathbb{R}^d \to \mathbb{R}^n$. Then $G$ is a width-$d$ feedforward network in $\mathbb{R}^d$ with monotonic activations, and $F|_H = A_L \circ G$.

Suppose for contradiction that $F(X), F(Y) \subset \mathbb{R}^n$ are linearly separable by a hyperplane $\pi^\top z = \alpha$. Pulling back through $A_L$, the linear functional $(A_L^\top \pi)$ on $\mathbb{R}^d$ satisfies $(A_L^\top \pi)^\top G(x) + (\pi^\top A_L(0) - \alpha) = \pi^\top F(x) - \alpha$, which is positive on $G(X)$ and negative on $G(Y)$. Hence $G(X)$ and $G(Y)$ are linearly separable in $\mathbb{R}^d$. But $\text{link}(X, Y) \neq 0$, so by Theorem 4.7 no width-$d$ feedforward network with monotonic activations can render $X, Y \subset \mathbb{R}^d$ linearly separable, a contradiction. $\square$

The bottleneck dimension $d$ alone determines the topological constraint; the input dimension $n$ is irrelevant. To break the obstruction one must widen the bottleneck or place full-width nonmonotonic/skip-augmented layers before compression.

## A.2. Classifier Heads and Linear Separability

A standard classifier head maps features $z \in \mathbb{R}^d$ to logits $\ell = W z + b \in \mathbb{R}^c$, predicting $\hat{y} = \arg\max_i \ell_i$. The class-$i$ decision region $D_i = \{z \in \mathbb{R}^c : z_i > z_j \text{ for all } j \neq i\}$ is the intersection of $c - 1$ open half-spaces, hence convex, and the $\{D_i\}$ are pairwise disjoint.

**Proposition A.6** (Binary classification equivalence). *For $c = 2$, $\hat{y} = \arg\max\{\ell_0, \ell_1\} = \mathbf{1}\big[(w_0 - w_1)^\top z + (b_0 - b_1) > 0\big]$. Two classes are perfectly classified by such a head iff their feature representations are linearly separable.*

Hence proving that a width-$d$ network cannot separate linked components is equivalent to proving that no classifier head achieves perfect accuracy on them.

# B. Invertible Architectures and Ambient Homeomorphisms

An *ambient homeomorphism* of $\mathbb{R}^d$ is a continuous bijection $h : \mathbb{R}^d \to \mathbb{R}^d$ with continuous inverse. By the invariance of domain (Brouwer 1912; (Munkres, 2000, Thm. 36.5)), any continuous injective map $f : U \to \mathbb{R}^d$ on an open $U \subseteq \mathbb{R}^d$ is an open map onto its image, hence a homeomorphism onto $f(U)$, though not in general an *ambient* homeomorphism of $\mathbb{R}^d$. The architectures we analyze in this appendix (flow-based models, Neural ODEs, normalizing flows) are constructed to be ambient homeomorphisms by design: their forward maps are explicit continuous bijections $\mathbb{R}^d \to \mathbb{R}^d$ with continuous inverses. For such architectures, the lemmas below apply globally; for a continuous injective width-$d$ feedforward network, they apply on the image of the data manifold under that network, which suffices for the linking-preservation conclusion.

**Lemma B.1** (Ambient homeomorphisms preserve component count and intrinsic invariants). *For disjoint compact connected sets $X_1, \ldots, X_k \subset \mathbb{R}^n$ and any ambient homeomorphism $h : \mathbb{R}^n \to \mathbb{R}^n$, the images $h(X_1), \ldots, h(X_k)$ are $k$ pairwise disjoint compact connected sets, and every intrinsic topological invariant of each $X_i$ (homotopy type, homology, knot complement type, etc.) is preserved.*

*Proof.* Continuity preserves connectedness, bijectivity preserves disjointness, and $h$ restricted to each $X_i$ is a homeomorphism onto $h(X_i)$, so every homotopy- or homeomorphism-invariant of $X_i$ transfers. This justifies the "cannot merge connected components" and "cannot fill holes" rows of the AH (ambient homeomorphism) column in Table 1; the "cannot unlink" row uses the companion linking-preservation result (Lemma B.2). $\qquad\square$

**Lemma B.2** (Ambient homeomorphisms preserve linking numbers). *For disjoint simple closed curves $X, Y \subset \mathbb{R}^3$ and any ambient homeomorphism $h : \mathbb{R}^3 \to \mathbb{R}^3$, $\mathrm{link}(h(X), h(Y)) = \pm \mathrm{link}(X, Y)$ (+ if $h$ is orientation-preserving, − otherwise).*

*Proof.* The linking number is the degree of the Gauss map $\Gamma : X \times Y \to \mathbb{S}^2$ (Definition C.1); $h$ induces a self-map of $X \times Y$ preserving degree up to the orientation sign. $\qquad\square$

### B.1. Invertible Architectures in Machine Learning

Invertible architectures whose forward map is an ambient homeomorphism therefore cannot change linking number at any depth. This includes reversible residual networks (RevNet (Gomez et al., 2017), i-RevNet (Jacobsen et al., 2018)) and normalizing flows (Dinh et al., 2017; Kingma & Dhariwal, 2018), which are invertible by construction; and ODE-based flows (Neural ODEs (Chen et al., 2018), FFJORD (Grathwohl et al., 2019), Flow Matching (Lipman et al., 2023)), whose ODE integration of a Lipschitz vector field produces a diffeomorphism. Dupont et al. (2019) verify this constraint empirically on concentric-ring classification with Neural ODEs; their augmented variant works precisely by lifting to higher dimension (width expansion). Discrete-time ResNets are *not* subject to this constraint: their layer-by-layer composition need not be a homeomorphism, and a discrete block can implement $|x| = x + 2\,\mathrm{ReLU}(-x)$ (Theorem 5.2), which is not invertible. One-step direct evaluators, e.g., MeanFlow (Geng et al., 2025), similarly escape the homeomorphism constraint by avoiding ODE integration entirely.

## C. Proofs for Sections 3 and 4: Linking-Number Preservation

This appendix proves the main linking-preservation and impossibility theorems for both the $\mathbb{R}^3$/ReLU setting of Section 3 and the general width-$d$/coordinate-wise-monotonic setting of Section 4. The general results subsume the $\mathbb{R}^3$/ReLU special case; we present the general proofs and indicate where the elementary $\mathbb{R}^3$ argument specializes.

### C.1. Linking number and link homotopy

**Definition C.1** (Link and linking number in $\mathbb{R}^3$). A *link* is a finite collection of disjoint simple closed curves in $\mathbb{R}^3$. For disjoint oriented simple closed curves $X, Y \subset \mathbb{R}^3$ the *linking number* $\mathrm{link}(X, Y) \in \mathbb{Z}$ is the Gauss integral

$$\mathrm{link}(X, Y) = \frac{1}{4\pi} \oint_X \oint_Y \frac{(x - y) \cdot (dx \times dy)}{|x - y|^3},$$

equivalently the signed crossing-count $\frac{1}{2} \sum_{p \in \pi(X) \cap \pi(Y)} \epsilon_p$ for any regular projection $\pi : \mathbb{R}^3 \to \mathbb{R}^2$.

**Definition C.2** (Degree and higher-dimensional linking number). The *degree* of a continuous map $G : X \to \mathbb{S}^k$ from a closed oriented $k$-manifold $X$ is $\deg(G) = \sum_{x \in G^{-1}(y)} \mathrm{sign}(\det DG_x)$ at any regular value $y \in \mathbb{S}^k$. For disjoint closed oriented manifolds $M^m, N^n \subset \mathbb{R}^d$ with $d = m + n + 1$, the *linking number* $\mathrm{link}(M, N)$ is the degree of the Gauss map $G(x, y) = (x - y)/|x - y| : M \times N \to \mathbb{S}^{d-1}$; for $m = n = 1$, $d = 3$ this recovers Definition C.1.

**Definition C.3** (Link homotopy). A *link homotopy* of disjoint compact subsets $X_1, \ldots, X_k \subset \mathbb{R}^n$ is a continuous map $H : \bigsqcup_i X_i \times [0, 1] \to \mathbb{R}^n$ with $H(\cdot, 0)$ the inclusion and $H(X_i, t) \cap H(X_j, t) = \varnothing$ for all $t \in [0, 1]$ and $i \neq j$. Two configurations are *link homotopic* if connected by such a homotopy.

*Remark* C.4 (Parametrized-image convention). In the layerwise arguments below, notation such as $X^{(j)}$ and $Y^{(j)}$ denotes the images of the composed maps

$$f_j = F_j \circ \cdots \circ F_1 \circ \iota_X : M \to \mathbb{R}^d, \qquad g_j = F_j \circ \cdots \circ F_1 \circ \iota_Y : N \to \mathbb{R}^d,$$

with the parametrizing maps left implicit. Thus $\text{link}(X^{(j)}, Y^{(j)})$ means the Gauss map degree of

$$(u, v) \mapsto \frac{f_j(u) - g_j(v)}{\|f_j(u) - g_j(v)\|},$$

which is well-defined whenever $f_j(M) \cap g_j(N) = \varnothing$, even if one component has self-intersections or the composed maps are not embeddings (Rolfsen, 1976, Ch. 5); see also (Hatcher, 2002, Sec. 2.2). A link homotopy is likewise a homotopy of these maps, equivalently $h_t \circ f_j$ and $h_t \circ g_j$ when written using an ambient homotopy $h_t : \mathbb{R}^d \to \mathbb{R}^d$. We keep the set notation as a standard abuse of notation, with the underlying composed maps understood.

**Lemma C.5** (Link homotopy invariance). *If $(M', N')$ is link homotopic to $(M, N)$ in $\mathbb{R}^d$ ($d = m + n + 1$), then $\text{link}(M', N') = \text{link}(M, N)$.*

*Proof.* The Gauss map varies continuously with $t$ along the link homotopy (disjointness keeps it well-defined), and degree is a homotopy invariant; equivalently, $t \mapsto \text{link}(H_M(M, t), H_N(N, t))$ is a continuous $\mathbb{Z}$-valued function on $[0, 1]$, hence constant. $\square$

**Lemma C.6** (Linear separability implies $\text{link} = 0$). *If two disjoint closed oriented manifolds $M^m, N^n \subset \mathbb{R}^d$ ($d = m+n+1$) are linearly separable, then $\text{link}(M, N) = 0$.*

*Proof.* A separating hyperplane partitions $\mathbb{R}^d$ into disjoint open half-spaces $H^+ \ni M$ and $H^- \ni N$. Straight-line contractions of $M$ to a point $p \in H^+$ and $N$ to a point $q \in H^-$ stay in their respective half-spaces and so remain disjoint, defining a link homotopy. At the endpoint of the homotopy, the Gauss map $M \times N \to \mathbb{S}^{d-1}$ is the constant map sending $(x, y)$ to $(p - q)/|p - q|$, hence has degree zero; by Lemma C.5 this degree equals $\text{link}(M, N)$, so $\text{link}(M, N) = 0$. $\square$

## C.2. Preservation under monotonic activations

**Lemma C.7** (Monotonic-activation linking preservation). *Let $\sigma : \mathbb{R}^d \to \mathbb{R}^d$ be coordinate-wise monotonic, with $r$ nonincreasing coordinates. If disjoint compact manifolds $M, N \subset \mathbb{R}^d$ satisfy $\sigma(M) \cap \sigma(N) = \varnothing$, then*

$$\text{link}(\sigma(M), \sigma(N)) = (-1)^r \text{link}(M, N).$$

*Proof.* Decompose $\sigma = R \circ \sigma'$ coordinate-wise as follows. For each coordinate $i$, set $\sigma'_i = \sigma_i$ if $\sigma_i$ is nondecreasing and $\sigma'_i = -\sigma_i$ if $\sigma_i$ is nonincreasing; then every $\sigma'_i$ is nondecreasing. Let $R : \mathbb{R}^d \to \mathbb{R}^d$ be the diagonal reflection that negates each output coordinate $i$ where $\sigma_i$ was nonincreasing ($R$ has $r_i = -1$ for those $r$ coordinates and $r_i = +1$ for the rest). Then $R_i(\sigma'_i(t)) = \sigma_i(t)$ for every $i$ and every $t$, so $\sigma = R \circ \sigma'$.

*Stage 1: link homotopy via $\sigma'$.* The straight-line interpolation $H_t(x) = (1 - t)x + t\sigma'(x)$, applied simultaneously to $M$ and $N$, is a link homotopy. Suppose $H_t(x) = H_t(y)$ with $x \in M$, $y \in N$. Pick coordinate $i$ with $x_i \neq y_i$ (exists since $M \cap N = \varnothing$); WLOG $x_i < y_i$. Since $\sigma'_i$ is nondecreasing, $\sigma'_i(x_i) \leq \sigma'_i(y_i)$, so $H_t(x)_i = (1 - t)x_i + t\sigma'_i(x_i) < (1 - t)y_i + t\sigma'_i(y_i) = H_t(y)_i$ for $t \in [0, 1)$. At $t = 1$ equality would require $\sigma'(x) = \sigma'(y)$; applying $R$ gives $\sigma(x) = \sigma(y)$, contradicting $\sigma(M) \cap \sigma(N) = \varnothing$. By Lemma C.5, $\text{link}(\sigma'(M), \sigma'(N)) = \text{link}(M, N)$.

*Stage 2: reflection.* $R$ is a homeomorphism of $\mathbb{R}^d$ with $\det R = (-1)^r$. Under $R$, the Gauss map $G : M \times N \to \mathbb{S}^{d-1}$ (Definition C.2) for any disjoint $M, N \subset \mathbb{R}^d$ becomes $G_R(x, y) = R(x - y)/|R(x - y)|$; since $R$ is an isometry of $\mathbb{R}^d$ restricted to a degree-$(-1)^r$ self-map of $\mathbb{S}^{d-1}$, the degree of the Gauss map is multiplied by $(-1)^r$. Hence $\text{link}(R(\sigma'(M)), R(\sigma'(N))) = (-1)^r \text{link}(\sigma'(M), \sigma'(N)) = (-1)^r \text{link}(M, N)$, as claimed. $\square$

For the $\mathbb{R}^3$/ReLU special case with curves $X, Y$, the same conclusion admits an elementary one-coordinate-at-a-time argument: write $\sigma = G_3 \circ G_2 \circ G_1$ where $G_j$ applies ReLU only to coordinate $j$, and homotope $X$ then $Y$ through each $G_j$ in turn. A collision $H_t^X(x) = y$ would force $G_j(x) = G_j(y)$ at the endpoint, contradicting $\sigma(X) \cap \sigma(Y) = \varnothing$. We refer to this elementary form below as the *sequential ReLU homotopy*.

## C.3. Rank-deficient transformations force intersection

**Lemma C.8** (Rank-deficient intersection, general $\mathbb{R}^d$). *Let $M^m, N^n \subset \mathbb{R}^d$ be disjoint closed oriented submanifolds with $m + n + 1 = d$ and $\text{link}(M, N) \neq 0$. For any rank-deficient affine $f : \mathbb{R}^d \to \mathbb{R}^d$, $f(M) \cap f(N) \neq \varnothing$.*

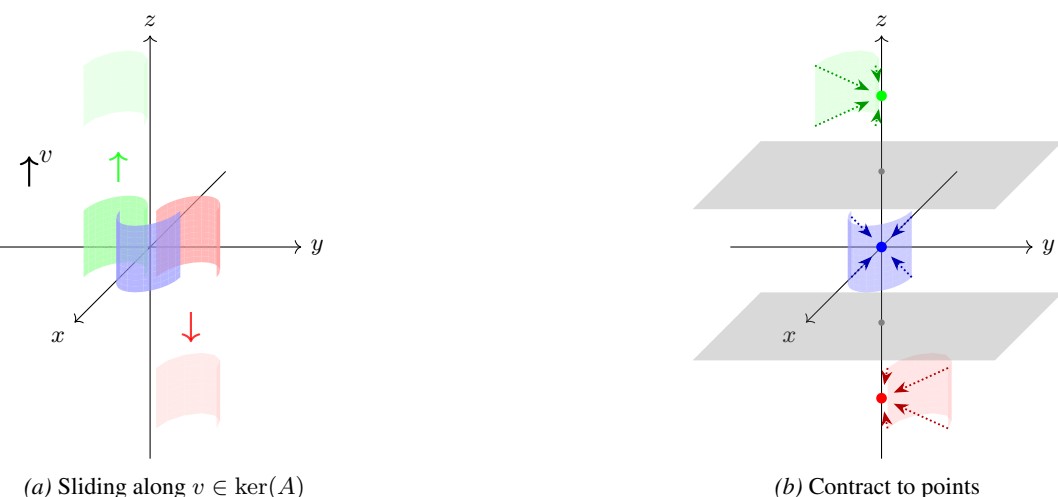

*(a)* Sliding along $v \in \ker(A)$        *(b)* Contract to points

*Figure 8.* Kernel translation argument. If a rank-deficient affine map with $v$ in its kernel does not create intersections, we can slide components along $v$ (unchanged by $f$) to achieve linear separation, then contract each to a point, forcing $\mathrm{link} = 0$.

*Proof.* Suppose not. Write $f(x) = Ax + b$ with $\mathrm{rank}(A) < d$ and pick a unit vector $v \in \ker(A)$. Since $M, N$ are compact, choose $L$ large enough that $M$ and $N + Lv$ lie in disjoint open half-spaces normal to $v$ (hence are linearly separable).

Translation along $v$ commutes with $f$: $f(N + tv) = AN + Atv + b = AN + b = f(N)$ for every $t$, so $f(M) \cap f(N + tv) = f(M) \cap f(N) = \varnothing$ for all $t \in [0, L]$. Hence any putative collision $x = y + tv$ during the translation would force $f(x) = f(y)$, contradicting $f(M) \cap f(N) = \varnothing$. So $t \mapsto (M, N + tv)$ is a link homotopy. By Lemma C.5, $\mathrm{link}(M, N + Lv) = \mathrm{link}(M, N) \neq 0$. But $(M, N + Lv)$ is linearly separable, so Lemma C.6 gives $\mathrm{link}(M, N + Lv) = 0$, a contradiction. $\square$

In $\mathbb{R}^3$, every rank-deficient affine map factors through a rank-2 orthogonal projection $P : \mathbb{R}^3 \to \mathbb{R}^2$; the combinatorial linking formula then equates $\mathrm{link}(X, Y)$ with a signed sum over $P(X) \cap P(Y)$, so non-zero linking forces a non-empty projection intersection. This is the special case used to prove Theorem 3.7.

### C.4. Main impossibility theorems

*Proof of Theorem 4.7 (general impossibility).* Suppose a width-$d$ feedforward network $F = A_L \circ \sigma_{L-1} \circ A_{L-1} \circ \cdots \circ \sigma_1 \circ A_1$ with coordinate-wise monotonic $\sigma_i$ achieves linear separability of $M, N \subset \mathbb{R}^d$ with $\mathrm{link}(M, N) \neq 0$. Here $M^{(j)}, N^{(j)}$ are understood in the parametrized-image sense of Remark C.4. Linear separability forces $M^{(j)} \cap N^{(j)} = \varnothing$ at every layer (else collision propagates). We prove by induction on $j$ that $\mathrm{link}(M^{(j)}, N^{(j)}) = \pm \mathrm{link}(M, N) \neq 0$: invertible affine layers preserve $\mathrm{link}$ up to sign (ambient homeomorphisms preserve degree); monotonic activations preserve $\mathrm{link}$ up to sign by Lemma C.7; rank-deficient affine layers would create an intersection by Lemma C.8, contradicting disjointness. Therefore $\mathrm{link}(F(M), F(N)) = \pm \mathrm{link}(M, N) \neq 0$. But linear separability requires $\mathrm{link} = 0$ by Lemma C.6. Contradiction. $\square$

*Proof of Theorem 3.7 ($\mathbb{R}^3$/ReLU).* The same argument with $d = 3$ and $\sigma_i = \mathrm{ReLU}$; the sequential ReLU homotopy substitutes for Lemma C.7. $\square$

*Proof of Theorem 4.5 (higher-dimensional).* The same argument with general $d = m + n + 1$; Lemma C.6 and Lemma C.7 apply verbatim. $\square$

## D. Width Upper Bound for Topological Unlinking

This section shows that width $d + 1$ is sufficient to eliminate the *classification* obstruction: a width-$(d + 1)$ ReLU network can linearly separate any disjoint compact configuration in $\mathbb{R}^d$ regardless of linking. The construction does not unlink the classes as ambient subsets of $\mathbb{R}^d$; it maps them into disjoint scalar intervals, which is the operation classification actually requires.

**Theorem D.1** (Unlinking via width-$(d+1)$ networks). *Let $X_1, \dots, X_k \subset \mathbb{R}^d$ be disjoint compact subsets in any linking configuration. For any $\varepsilon \in (0, 1/2)$, there exists a feedforward network $F : \mathbb{R}^d \to \mathbb{R}$ with ReLU activations and width $d+1$ such that $|F(x) - i| < \varepsilon$ for every $x \in X_i$, so the images $F(X_i) \subset (i - \frac{1}{2}, i + \frac{1}{2})$ lie in disjoint convex intervals.*

*Proof.* Disjoint compact sets in $\mathbb{R}^d$ admit disjoint open neighborhoods $U_i \supset X_i$ together with Urysohn cutoffs $\psi_i : \mathbb{R}^d \to [0, 1]$ with $\psi_i \equiv 1$ on $X_i$ and $\psi_i \equiv 0$ outside $U_i$. The target $\tilde{f}(x) = \sum_{i=1}^k i \cdot \psi_i(x)$ is continuous and equals $i$ on $X_i$. By the Hanin–Sellke width-$(d+1)$ universal approximation theorem (Hanin & Sellke, 2017), $\tilde{f}$ can be uniformly $\varepsilon$-approximated on the compact set $\bigcup_i \overline{U_i}$ by a width-$(d+1)$ ReLU network $F$. Then $|F(x) - i| < \varepsilon < \frac{1}{2}$ on each $X_i$, separating the images into disjoint intervals. To keep input and output dimensions equal, extend $F$ to $G(x) = (F(x), 0, \dots, 0)$. $\square$

Width $d+1$ is therefore tight: width $d$ is impossible by the lower bound below, while width $d+1$ suffices.

# E. Width Lower Bound for Universal Approximation

**Theorem E.1** (Width lower bound for universal approximation). *For any continuous coordinate-wise monotonic activation (ReLU, leaky-ReLU, sigmoid, tanh, etc.), the minimum width $w_{\min}$ for which width-$w_{\min}$ feedforward networks $F : \mathbb{R}^d \to \mathbb{R}$ are uniform universal approximators on compact sets satisfies $w_{\min} \geq d+1$.*

*Proof.* Suppose for contradiction that width-$d$ networks with some monotonic $\sigma$ are dense in $C(K)$ for every compact $K \subset \mathbb{R}^d$. We treat the three cases $d = 1$, $d = 2$, $d \geq 3$ in turn.

$d = 1$. A width-1 feedforward network is a composition of monotonic scalar maps (affine $\mathbb{R} \to \mathbb{R}$ composed with monotonic $\sigma : \mathbb{R} \to \mathbb{R}$); such a composition is itself monotonic. Monotonic functions on $\mathbb{R}$ are not dense in $C([0, 1])$ (they cannot approximate any nonmonotonic continuous function), contradicting universal approximation. Hence $w_{\min} \geq 2$.

$d = 2$. Take $M = \{p_0\}$ a point and $N$ a smooth simple closed curve in $\mathbb{R}^2$ encircling $p_0$ once, so $\mathrm{link}(M, N) = 1$ (the winding number / degree of the Gauss map $M \times N \to \mathbb{S}^1$). The function $f|_M = 0$, $f|_N = 1$ extends continuously to all of $\mathbb{R}^2$ by Tietze extension; a width-2 uniform approximation on the compact $M \cup N$ would give a linear separator, contradicting Theorem 4.7 with $(m, n, d) = (0, 1, 2)$.

$d \geq 3$. Write $d = m + n + 1$ with $m, n \geq 1$ and take a non-trivially linked pair $(M^m, N^n) \subset \mathbb{R}^d$ with $\mathrm{link}(M, N) = 1$: the Hopf link for $d = 3$, the linked-spheres construction of Appendix G.4 for $d > 3$. Same Tietze + Theorem 4.7 argument applies.

In all three cases, $w_{\min} \geq d+1$. $\square$

The contribution here is the proof technique, not the bound itself. Prior width-$d$ insufficiency results constrain either the activation class or its regularity: Hanin & Sellke (2017) (ReLU, level-set components), Johnson (2019) (activations approximable by injections, level-set topology), Park et al. (2021); Cai (2023); Li et al. (2023); Kim et al. (2024) (ReLU, leaky-ReLU, and compact-domain minimum-width analyses), and Rochau et al. (2024) (monotone *Lipschitz* activations, approximation-theoretic). Our argument requires only *continuous coordinate-wise monotonic* activations, with no Lipschitz regularity, smoothness, or approximation by injections, and obtains the bound as a direct corollary of ambient topological invariance, complementing the width-$(d+1)$ upper bound of Appendix D.

# F. Detailed Constructions Breaking Topological Constraints

## F.1. Why nonmonotonic activations break homotopy preservation

The monotonic-activation preservation lemma (Lemma C.7) used the straight-line homotopy $H_t(x) = (1 - t)x + t\sigma(x)$, which is a link homotopy precisely because monotonicity rules out coordinate-wise collisions during interpolation. For $\sigma(x) = |x|$, the same straight-line interpolation crosses the fold line $x_i = 0$ from negative to positive sides: distinct points with mirror-image coordinates can collide on the fold. Thus $\mathrm{link}(\sigma(M), \sigma(N))$ need not equal $\mathrm{link}(M, N)$, and the rigidity that drives the impossibility theorems disappears.

## F.2. Hopf link unlinking via absolute value activations

We give an explicit five-step construction (Figure 9) that takes the Hopf link $X(t) = (\cos t, \sin t, 0)$, $Y(s) = (0, 1 + \cos s, \sin s)$ (the same Hopf link as in Example 3.1, with the $X$-axis offset of the main text swapped to the $Y$-axis by a $90°$ rotation of coordinates, chosen so the first fold acts on $y$) to a linearly separable configuration in $\mathbb{R}^3$ using only coordinate-wise $|\cdot|$ and affine maps:

1. Apply $|\cdot|$ to the $y$-coordinate: $X \mapsto (\cos t, |\sin t|, 0)$.

2. Affine $(x, y, z) \mapsto (x, 1 - y, z)$: now $X = (\cos t, 1 - |\sin t|, 0)$ and $Y = (0, -\cos s, \sin s)$.

3. Apply $|\cdot|$ to the $x$- and $y$-coordinates.

4. Apply $|\cdot|$ to the $z$-coordinate.

After step 4, both curves lie in the positive octant; an explicit hyperplane (shaded in Figure 9) separates them.

**Extension to activations with a local extremum.** The recipe generalizes to any activation $\sigma : \mathbb{R} \to \mathbb{R}$ with a strict local extremum on an open interval $I \subset \mathbb{R}$, e.g., the local minimum of GELU near $-0.5$, of Swish/SiLU near $-1.3$, of Mish near a similar point: since the data is compact, an affine pre-shift $x \mapsto ax + b$ rescales the relevant data coordinate into $I$, on which $\sigma|_I$ is nonmonotonic. The straight-line homotopy $t \mapsto (1 - t)x + t\sigma(x)$ on $I$ admits a fold-line collision in exactly the same way as $|\cdot|$ does on $\mathbb{R}$, so the same disjoint-tube argument fails and the same unlinking construction goes through with $\sigma|_I$ in place of $|\cdot|$. We do not require $\sigma|_I$ to equal $|\cdot|$, only to fold the rescaled data into a region with reduced crossing structure.

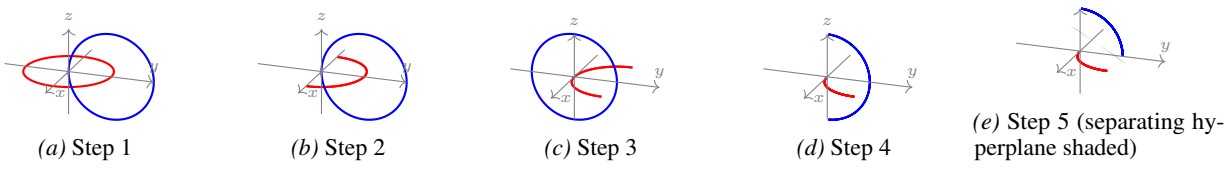

*(a)* Step 1      *(b)* Step 2      *(c)* Step 3      *(d)* Step 4      *(e)* Step 5 (separating hyperplane shaded)

*Figure 9.* Hopf link unlinking via absolute value activations.

## F.3. ResNet absolute-value synthesis

*Proof of Theorem 5.2 (ResNet topological expressivity).* The single identity

$$|x| = x + 2\,\text{ReLU}(-x)$$

realizes coordinate-wise absolute value as one ResNet block: take residual branch $\mathcal{G}(x) = 2\,\text{ReLU}(-x)$ (a width-$d$ ReLU sublayer with input weight $-I_d$ and output weight $2I_d$), then $\mathcal{F}(x) = x + \mathcal{G}(x) = |x|$. Translated folds $x \mapsto c + |x - c|$ follow by composing affine shifts. Iterating coordinate-wise yields the unlinking construction of Figure 9, so a width-$d$ ReLU ResNet inherits the topological expressivity of absolute-value activations. $\square$

## F.4. Transformer absolute-value synthesis

*Proof of Theorem 5.3 (transformer topological expressivity).* Process each coordinate $x_i$ independently as a two-token sequence $(x_i, x_i)$ with distinct positional encodings $(p_1, p_2)$. With scalar query/key/value/output weights and biases $(w_q, w_k, w_v, w_o, b_q, b_k, b_v, b_o)$, the second-position attention weight reduces to a sigmoid:

$$\alpha_{21} = \text{softmax}(q_2 k_1, q_2 k_2)_1 = \text{sigmoid}\big(q_2(k_1 - k_2)\big),$$

giving output $g(x_i) = w_o\,[\alpha_{21}v_1 + (1 - \alpha_{21})v_2] + b_o$. Choosing for instance $w_q = -5$, $w_k = 5$, $w_v = w_o = 1$, $p_1 = 0$, $p_2 = 1$, $b_q = 4.3$, $b_k = b_v = b_o = 0$ produces a function with a local minimum near $x = 0$, decreasing for $x < 0$ and increasing for $x > 0$, a smoothed V-shape that locally resembles $|x|$ near the origin.

This construction is a per-coordinate *local* surrogate for $|\cdot|$, not a global approximation. For the unlinking sequence of Figure 9, an affine pre-shift first rescales the compact data into the V-shape's effective neighborhood; the resulting attention

map then implements a nonmonotonic coordinate fold close enough to $|\cdot|$ for the topological transformation, since the construction depends only on the existence of a coordinate-wise fold (any nonmonotonic surrogate with the required folding behavior on the rescaled data domain suffices), not on exact equality with $|\cdot|$. Composing this attention-fold step with the affine transformations from each step of the unlinking sequence gives a pure-attention realization of the same topological transformation, breaking the linking obstruction. $\qquad\square$

# G. Experimental Details

### G.1. Hopf Link Parametrization and Thickening

**Hopf link parametrization.** $X(t) = (\cos t, \sin t, 0)$ and $Y(s) = (1 + \cos s, \, 0, \, \sin s)$.

**Thickening procedure.** Sample points as $\gamma(t) + \varepsilon \cdot \mathbf{n}(t)$ where $\mathbf{n}(t)$ is a unit normal to the curve, $\varepsilon \sim \mathcal{U}(0, r)$ with $r = 0.15$, and high-frequency oscillations $0.3\sin(100t)$ are added to preserve topology.

### G.2. Network Architectures and Training Protocol

All architectures use width 3. FFNs are fully connected with depths 3–20; ReLU ResNets use residual blocks $x \mapsto x + g(x)$ with 2-layer width-3 subnetworks $g$. We train with Adam for FFNs and AdamW for ResNets at learning rate $10^{-3}$, batch size 128, up to 800 epochs, early stopping with patience 100–200, and cross-entropy loss on 6000 points (3000 per class) split 80/20 for train/validation. Code: github.com/7pocheR/low_dimensional_topology.

### G.3. ReLU vs GELU: Detailed Observations

Table 2 reveals three patterns. (i) Across 30 seeds, ReLU's best run never exceeds the $\sim$90% topological ceiling at any depth (max 92.8% at depth 3); mean accuracy is bounded above by the ceiling and additionally degrades with depth as optimization difficulty compounds the expressivity barrier, consistent with Theorem 3.7. (ii) GELU mean accuracy stays at 89–91% for depths 3–12 and the best run achieves 100% at depths 5–12, confirming that nonmonotonic activations escape the constraint (§5.1). (iii) At extreme depths ($\geq 16$) both activations suffer optimization instability (large stds); depth alone cannot compensate. Unlike standard universal-approximation results where depth substitutes for width, here depth provides no escape from the topological barrier.

### G.4. Higher-Dimensional Linked Spheres Construction

For the higher-dimensional experiments (Section 6.5), we construct two $n$-spheres linked in $\mathbb{R}^{2n+1}$ with linking number $\pm 1$. The construction uses explicit parametrizations $\tilde{A}, \tilde{B} : \mathbb{S}^n \to \mathbb{R}^{2n+1}$ derived from stereographic projection.

**Parametrization.** For $u = (u_0, u') \in \mathbb{S}^n \subset \mathbb{R}^{n+1}$ with $u' = (u_1, \ldots, u_n)$, set $a = 1 - u_0/\sqrt{2}$ and define $\tilde{A}(u) = (u'/a, -u_0/(\sqrt{2}\,a), \mathbf{0}_n) \in \mathbb{R}^n \times \mathbb{R} \times \mathbb{R}^n = \mathbb{R}^{2n+1}$, where $\mathbf{0}_n$ is the zero vector in $\mathbb{R}^n$. Symmetrically, for $v = (v_0, v') \in \mathbb{S}^n$ set $b = 1 - v_0/\sqrt{2}$ and define $\tilde{B}(v) = (\mathbf{0}_n, v_0/(\sqrt{2}\,b), v'/b)$. The images $\tilde{A}(\mathbb{S}^n), \tilde{B}(\mathbb{S}^n)$ are disjoint $n$-spheres in $\mathbb{R}^{2n+1}$ with $\mathrm{link}(\tilde{A}(\mathbb{S}^n), \tilde{B}(\mathbb{S}^n)) = 1$ (verified numerically via the higher-dimensional Gauss linking integral).

**Geometric structure.** The embedding places $\tilde{A}(\mathbb{S}^n)$ in the $X$-$Z$ subspace (first $n$ coordinates plus middle coordinate, with $Y = \mathbf{0}$) and $\tilde{B}(\mathbb{S}^n)$ in the $Z$-$Y$ subspace (middle coordinate plus last $n$ coordinates, with $X = \mathbf{0}$). These coordinate subspaces intersect only along the shared middle coordinate axis, and the linking arises from the spheres' interlocking configuration around this axis.

**Minimum separation.** The minimum distance between points on the two spheres is $d_{\min} = 2(\sqrt{2} - 1) \approx 0.828$, independent of $n$. This ensures the spheres remain well-separated.

**Targeted thickening.** To create training data with non-trivial volume, we use *targeted thickening*: each sphere is thickened within its complementary (normal) subspace to preserve the linking structure. In the $(X, Z, Y)$ split with $X, Y \in \mathbb{R}^n$, $Z \in \mathbb{R}$: $\tilde{A}$ lies in the $X$-$Z$ subspace ($Y = 0$) and is thickened in the $Y$-direction as $\tilde{A}_\rho(u, \eta) = \tilde{A}(u) + (0, 0, \eta)$; $\tilde{B}$ lies in the $Z$-$Y$ subspace ($X = 0$) and is thickened in the $X$-direction as $\tilde{B}_\rho(v, \zeta) = \tilde{B}(v) + (\zeta, 0, 0)$, with $\eta, \zeta \sim \mathrm{Uniform}(B_n(\rho))$ on the $n$-dimensional radius-$\rho$ ball. Our experiments use $\rho = 0.5$.

**Multi-copy placement.** To amplify the topological barrier (Section 6.5), we place $k$ disjoint copies of the linked pair using $L^1$-*ordered grid placement*: copy centers are integer lattice points in $\mathbb{Z}^{2n+1}$ enumerated in nondecreasing $L^1$ norm order,

scaled by spacing $s = 10$. Specifically, we enumerate shells $\{v \in \mathbb{Z}^{2n+1} : \|v\|_1 = m\}$ for $m = 0, 1, 2, \ldots$ and take the first $k$ vectors. This spreads copies isotropically rather than along a single axis, preventing networks from exploiting directional biases. Each copy contributes an independent entanglement region, so networks must overcome $k$ local obstructions simultaneously.

### G.5. Linking Scaling Experiments: Variance and Dimensional Effect

For the higher-dimensional linking experiments (Table 4, $\mathbb{S}^2 \sqcup \mathbb{S}^2$ in $\mathbb{R}^5$ with $k$ disjoint copies), we tracked accuracy variance across 100 seeds in addition to best-case ceilings. At small $k$, ReLU+Skip is both the most reliable and reaches the best ceiling: at $k = 1$ it attains $100\%$ best and a $97.0 \pm 3.5$ mean test accuracy, whereas plain ReLU's mean is $89.4 \pm 11.4$ with seeds occasionally collapsing far below the topological ceiling. As $k$ grows, monotonic ceilings (ReLU, ReLU+Skip) decline faster than nonmonotonic ceilings (GELU, Swish): the nonmonotonic-minus-monotonic best-case gap is mildly negative for $k \le 5$, becoming $+3.7, +4.4, +3.9$pp at $k = 10, 20, 50$. The pattern is consistent with skip connections resolving the single entanglement at $k = 1$, while nonmonotonic activations are required to resolve each of many local entanglements at larger $k$. Mean accuracies decline for all architectures as $k$ grows, but nonmonotonic mean accuracies remain 4–6pp above monotonic at $k \ge 10$, e.g., GELU mean $81.0 \pm 3.7$ vs. ReLU+Skip mean $76.4 \pm 3.9$ at $k = 10$.

**Why higher dimensions need larger** $k$. At $n = 1$ ($\mathbb{S}^1 \sqcup \mathbb{S}^1$ in $\mathbb{R}^3$) the topological advantage of skip/nonmonotonic architectures is visible at $k = 1$ (Table 2); at $n = 2$ ($\mathbb{S}^2 \sqcup \mathbb{S}^2$ in $\mathbb{R}^5$) one needs $k > 5$ to see it. The reason is geometric: monotonic networks can achieve high accuracy by sacrificing accuracy on a small overlap region of characteristic size $\varepsilon$ near each linked location, whose volume fraction scales as $\Theta(\varepsilon^{2n+1})$ in ambient $\mathbb{R}^{2n+1}$. For $n = 1$ that fraction is $\sim 10\%$; for $n = 2$ the same $\varepsilon$ gives $\sim \varepsilon^2$ smaller (the $\sim 90\% \to \sim 99\%$ drop at $k = 1$ implies $\varepsilon \approx 0.3$). Real high-dimensional datasets are expected to compensate by having more linking opportunities (volume growing as $\Theta(\mathrm{poly}(n))$); our multi-copy design synthetically restores the cumulative obstruction so the experiment remains controlled across $n$.

### G.6. ResNet Skip Connection Visualization

Figure 10 shows the full resolution visualization of the ResNet skip connection mechanism on the disk-annulus ($\mathbb{S}^0$-$\mathbb{S}^1$) separation task. This experiment demonstrates that a depth-3 width-2 ResNet with ReLU activations learns to implement the folding operation $|x| = x + 2\,\mathrm{ReLU}(-x)$ predicted by Theorem 5.2.

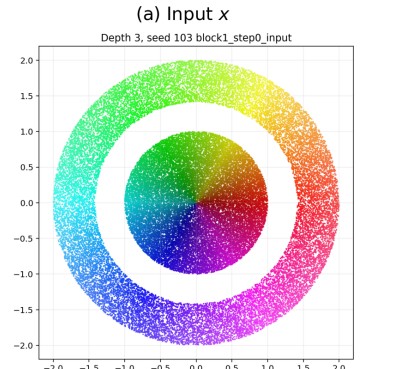 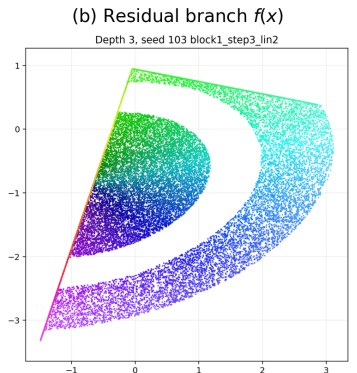 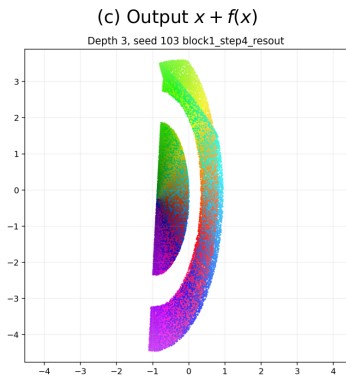

*Figure 10.* **Full resolution ResNet skip connection visualization.** (a) **Input** $x$: The nested disk (inner) and annulus (outer) form an $\mathbb{S}^0$-$\mathbb{S}^1$ link with link $= \pm 1$. Each point is assigned a color based on its angular position in the input space; this color is preserved across all three panels, allowing one to track where each input point is mapped by the network. Since colors vary continuously with position, a point's trajectory can be identified by locating the dot (or small patch of similar color) in each panel. (b) **Residual branch output** $f(x)$: The learned transformation approximates $2\,\mathrm{ReLU}(-x)$, "folding" negative coordinate values toward zero. Note the characteristic triangular shape created by the ReLU. (c) **Skip connection output** $x + f(x)$: Adding the residual to the input implements $|x|$, folding the configuration and separating the two components into vertically disjoint regions. The disk and annulus are now linearly separable. This experiment validates that ResNet can overcome topological barriers through the skip connection mechanism.

**Training details.** The ResNet architecture uses 3 residual blocks with width-2 hidden layers. Training uses Adam optimizer with learning rate 0.001 for 5000 epochs on 50,000 points per class. The visualization shows the block 1 transformation (input $\to$ first residual block output). Multiple random seeds (including seed 103 shown) consistently learn similar folding transformations, demonstrating that the theoretical mechanism is reliably discovered by gradient descent.

## G.7. Width Expansion Eliminates the Obstruction

Theorem D.1 predicts that increasing width past the critical $d + 1$ threshold removes the topological barrier. We verify this in $\mathbb{R}^7$ on $\mathbb{S}^3 \sqcup \mathbb{S}^3$ with $k = 10$ copies, depth 5, 15 seeds per width (Table 7). Critical width $d = 7$ caps at $\approx 88\%$ (mean) / 93% (max), and accuracy improves overall as width is expanded (with small finite-seed nonmonotonicities), saturating near $99-100\%$ around width $\approx 5d$. The same pattern holds in $\mathbb{R}^5$ ($k = 10$, depth 5): plain ReLU at width 20 already reaches 98.5% mean, and adding nonmonotonic activations or skip connections only marginally changes this, confirming that the architectural mechanisms (folding, skip) are most valuable when width is constrained below the critical threshold and become substitutable with width when width is relaxed.

*Table 7.* Width-expansion in $\mathbb{R}^7$ ($\mathbb{S}^3 \sqcup \mathbb{S}^3$, $k = 10$, depth 5, 15 ReLU seeds).

| Width | 7 | 8 | 10 | 14 | 21 | 28 | 35 | 49 |
|---|---|---|---|---|---|---|---|---|
| Multiplier of $d$ | $1\times$ | $1.1\times$ | $1.4\times$ | $2\times$ | $3\times$ | $4\times$ | $5\times$ | $7\times$ |
| Mean (%) | 87.9 | 86.7 | 91.2 | 95.4 | 98.8 | 99.4 | 99.6 | 99.7 |
| Max (%) | 93.1 | 93.9 | 94.8 | 98.0 | 99.2 | 99.7 | 100.0 | 99.9 |

## G.8. Layer-by-Layer Linking-Number and Min-Distance Tracking

To verify mechanistically that the impossibility theorem reflects what the network actually does, we track both the Gauss linking integral link and the minimum inter-class distance $d_{\min}$ layer-by-layer for a width-3, depth-5 network on the Hopf link, comparing ReLU, GELU, and ReLU+skip (Table 8).

*Table 8.* Layer-by-layer link and minimum inter-class distance $d_{\min}$ on the Hopf link (best seed, 200 points per class). Starred ReLU link values are artifacts (see Interpretation below).

| | Input | $L_0$ | $L_1$ | $L_2$ | $L_3$ | $L_4$ | $L_5$ |
|---|---|---|---|---|---|---|---|
| ReLU link | $-1$ | $0.50^*$ | $0.18^*$ | 0 | 0 | 0 | 0 |
| ReLU $d_{\min}$ | 0.83 | 0.06 | 0.00 | 0.00 | 0.00 | 0.00 | 0.00 |
| GELU link | $-1$ | 0 | 0 | 0 | 0 | 0 | 0 |
| GELU $d_{\min}$ | 0.83 | 0.11 | 0.17 | 0.63 | 1.01 | 1.32 | 1.41 |
| ReLU+skip link | $-1$ | 0 | 0 | 0 | 0 | 0 | 0 |
| ReLU+skip $d_{\min}$ | 0.83 | 0.38 | 0.46 | 1.05 | 2.21 | 3.04 | 3.73 |

**Interpretation.** ReLU forces the two curves into intersection by $L_1$ (minimum distance collapses to 0); the apparent fractional link values at $L_0$, $L_1$ are not meaningful linking numbers but artifacts of the Gauss integral becoming ill-conditioned as $|x - y| \to 0$ (linking number is only defined for disjoint curves). After $L_1$, the network has destroyed the geometric structure rather than resolved the topology, consistent with Lemma 3.5 and the impossibility theorem. GELU and ReLU+skip, by contrast, achieve link $= 0$ *while keeping the curves disjoint*: after the first unlinking layer ($d_{\min}$ small but positive), subsequent layers increase $d_{\min}$ steadily, implementing the genuine unlinking that monotonic feedforward networks cannot.

# H. Linking Detection Algorithm for Point Cloud Data

This section gives the algorithm used to detect topological linking between two finite point clouds $\mathcal{X}, \mathcal{Y} \subset \mathbb{R}^d$. The pipeline is: (i) project to $\mathbb{R}^3$ via PCA, (ii) build $k$-NN spatial graphs per class, (iii) extract a fundamental cycle basis per graph, (iv) compute Gauss linking numbers over pairs of basis cycles.

## H.1. Graph construction

For each class $\mathcal{X} = \{x_1, \ldots, x_n\} \subset \mathbb{R}^3$, the $\varepsilon$-*filtered $k$-NN graph* $G$ has vertex set $\{1, \ldots, n\}$ and edge $(i, j)$ whenever $j$ is among the $k$ nearest neighbors of $x_i$ and $\|x_i - x_j\| \leq \varepsilon$. The threshold $\varepsilon$ (typically a percentile of nearest-neighbor distances) suppresses spurious long edges. For fixed point cloud and $k$-NN relation, the edge set is monotone in $\varepsilon$: increasing $\varepsilon$ only adds eligible $k$-NN edges, so already detected witness cycles remain graph cycles at larger thresholds. The *mutual* variant additionally requires $i \in \mathrm{kNN}(j)$, yielding sparser but more symmetric graphs robust to density variation.

## H.2. Fundamental cycle basis via spanning forest

**Definition H.1** (Fundamental cycle basis). For a graph $G = (V, E)$ with $|V| = n$, $|E| = m$, and $c$ connected components, fix a spanning forest $T \subset E$ ($n - c$ edges). Each of the $m - n + c$ non-tree edges $e = (u, v)$ adds a unique cycle $C_e$ consisting of $e$ plus the unique $T$-path from $u$ to $v$. The collection $\{C_e\}$ is the *fundamental cycle basis*.

**Proposition H.2** (Cycle-basis sufficiency). *The fundamental cycle basis generates $H_1(G; \mathbb{Z})$, so every cycle is an integer linear combination of basis cycles. Linking number extends bilinearly to $H_1$, so $\mathcal{X}, \mathcal{Y}$ exhibit detectable linking iff some basis pair $(C_i, D_j)$ has $\mathrm{link}(C_i, D_j) \neq 0$.*

This reduces detection from a search over infinitely many cycle pairs to a finite computation over $O(\beta_{\mathcal{X}} \cdot \beta_{\mathcal{Y}})$ basis pairs, where $\beta = m - n + c$ is the first Betti number.

## H.3. Gauss linking number

For disjoint piecewise-linear cycles $C_1, C_2 \subset \mathbb{R}^3$, we evaluate the Gauss integral $\mathrm{link}(C_1, C_2) = \frac{1}{4\pi} \oint_{C_1} \oint_{C_2} \frac{(x-y) \cdot (dx \times dy)}{|x-y|^3}$ via midpoint quadrature on subdivided edges. Algorithm 1 packages the full detection pipeline.

---

**Algorithm 1** Link Detection for Point Cloud Data

---

**Require:** Point clouds $\mathcal{X}, \mathcal{Y} \subset \mathbb{R}^d$; parameters $k, \varepsilon$, subdivision $N_{\mathrm{sub}}$
**Ensure:** Linked decision + witness cycle pair if linked
 1: Project $\mathcal{X} \cup \mathcal{Y}$ to $\mathbb{R}^3$ via PCA
 2: Build $\varepsilon$-filtered $k$-NN graphs $G_{\mathcal{X}}, G_{\mathcal{Y}}$
 3: Compute spanning forests via BFS; extract fundamental cycle bases $\mathcal{C}_{\mathcal{X}}, \mathcal{C}_{\mathcal{Y}}$
 4: **if** $\mathcal{C}_{\mathcal{X}} = \varnothing$ or $\mathcal{C}_{\mathcal{Y}} = \varnothing$ **then**
 5:   **return** (Not linked: insufficient cycles)
 6: **end if**
 7: **for** $C \in \mathcal{C}_{\mathcal{X}}$, $D \in \mathcal{C}_{\mathcal{Y}}$ **do**
 8:   $\ell \leftarrow$ midpoint-quadrature Gauss integral over $C \times D$ with $N_{\mathrm{sub}}$ subdivisions, rounded to nearest integer
 9:   **if** $\ell \neq 0$ **then**
10:     **return** (Linked, witness $(C, D, \ell)$)
11:   **end if**
12: **end for**
13: **return** (Not linked, all $|\mathcal{C}_{\mathcal{X}}| \cdot |\mathcal{C}_{\mathcal{Y}}|$ basis pairs checked)

---

## H.4. Parameters and complexity

Typical choices: $k \in [6, 15]$ (denser $k$ exposes more cycles at the cost of compute), $\varepsilon$ at the 70th percentile of nearest-neighbor distances, $N_{\mathrm{sub}} \in \{4, 8\}$ subdivisions per edge, and minimum cycle length $\geq 4$ (smaller cycles are construction artifacts). Use mutual $k$-NN for heterogeneous density.

Complexity: $k$-NN graph in $O(n \log n)$ via $k$-d trees; spanning forest and cycle-basis extraction in $O(n + m) = O(kn)$; Gauss integral per cycle pair in $O(L_1 L_2 N_{\mathrm{sub}}^2)$ for cycle lengths $L_1, L_2$. Total: $O(n \log n + \beta_{\mathcal{X}} \beta_{\mathcal{Y}} L^2 N_{\mathrm{sub}}^2)$. For our CIFAR-10 setting ($n \sim 10^4$, $\beta \sim 10^2$, $L \sim 20$) the algorithm runs in tens of seconds per class pair.

# I. CIFAR-10 Linking Experiments: Details

This appendix gives the data-preparation, detection-parameter, and architectural details for the suggestive CIFAR-10 linking analysis of Section 6.6. We emphasize throughout that the evidence here is *correlational*, not causal, and rests on a 3D PCA projection rather than the native data manifold; the synthetic experiments (Appendix G) are the primary validation of the theory.

## I.1. Data preparation and link detection

**Augmentation and projection.** We apply $20\times$ training-set augmentation (random horizontal flip, random crop with 4-pixel padding, color jitter $\pm 0.2$ for brightness/contrast/saturation) to CIFAR-10 (yielding 1,050,000 samples; 105,000 per class),

flatten + standardize, and project to $\mathbb{R}^3$ via PCA on the combined augmented dataset.

**Detection parameters.** We run Algorithm 1 with $k = 15$ mutual nearest neighbors and minimum cycle length 30. For the displayed bird–deer witness, we separately binary-search the threshold $\varepsilon$ to locate the smallest scale at which any pair of class cycles links. At $\varepsilon = 0.0338$ (0.22% of the 3D bounding-box diagonal 15.555), the bird (58-vertex) and deer (40-vertex) classes form interlocked cycles with link $= -1$ (Gauss integral $-1.0036$); see Figure 11.

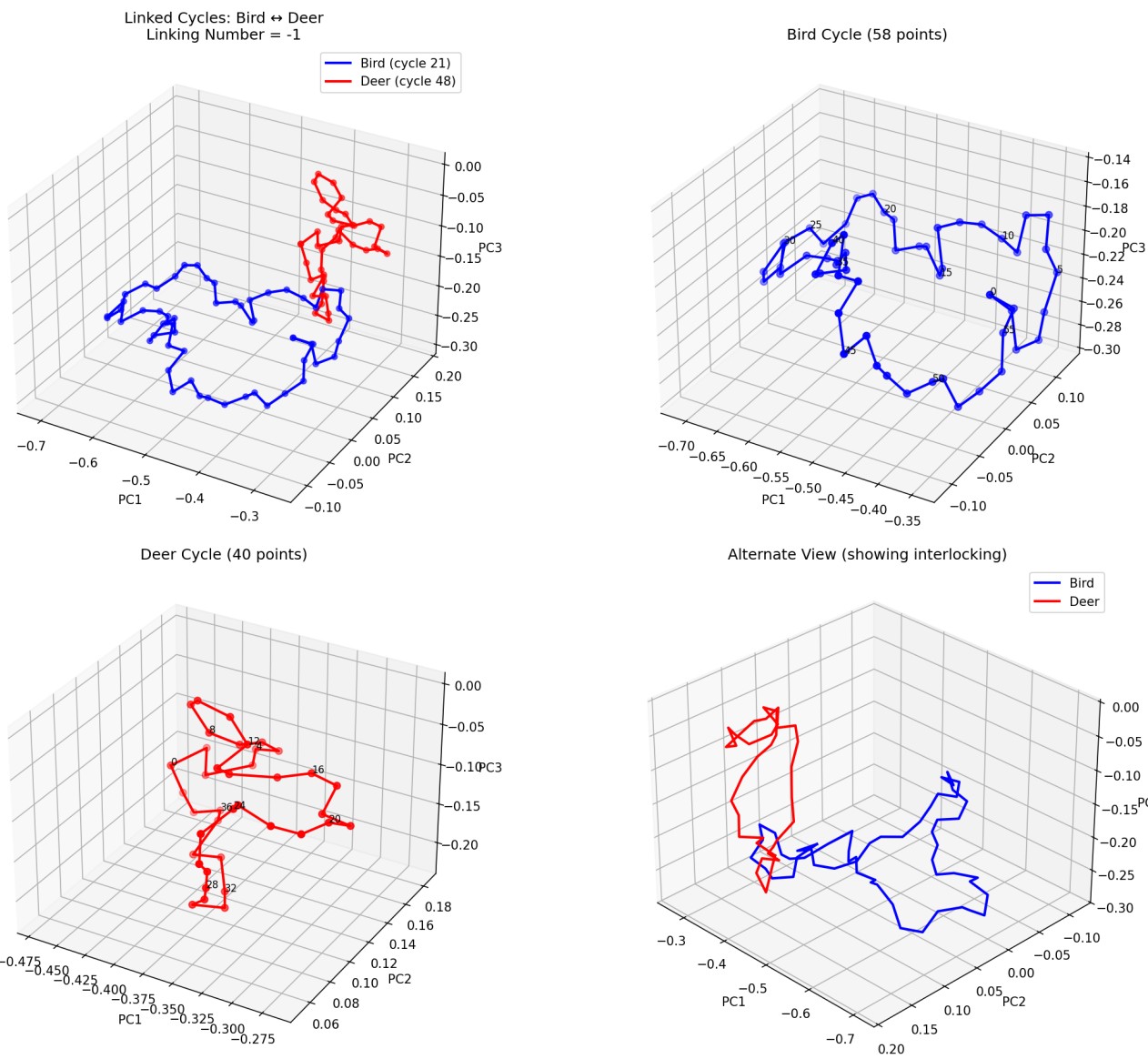

*Figure 11.* Linked cycles detected in CIFAR-10 PCA-3D: bird (blue, 58 points) and deer (red, 40 points) interlock with link $= -1$ at $\varepsilon = 0.034$.

**Linking-consistency definition.** For a class pair $(X_i, X_j)$ and $N$ independent runs (each regenerating the augmented dataset with a fresh random seed and recomputing graphs/cycles), the *linking consistency* is consistency$(X_i, X_j) = \frac{1}{N} \sum_r \mathbf{1}[|\operatorname{link}_r(X_i, X_j)| \geq 1]$. Variation across runs stems from augmentation randomness, sampling-dependent $k$-NN structure, and cycle-vertex placement; consistency measures how robustly a pair's linking survives sampling variability rather than being a property of the underlying manifold.

The consistency runs do *not* binary-search $\varepsilon$ independently for each pair. They use the same pre-arranged threshold sequence for all 45 pairs, $(0.11, 0.14, 0.15, 0.16, 0.17, 0.18, 0.19, 0.2, 0.3, 1.0, 30.0)$. Because the $\varepsilon$-filtered graph is monotone in $\varepsilon$,

a detected witness remains available at larger thresholds; the sequence therefore measures how small a scale suffices for detection, rather than whether some tuned threshold exists.

### I.2. 10-class linking consistency

Eleven runs across all $45$ CIFAR-10 class pairs give a deliberately coarse robustness measure rather than a definitive topological classification. The distribution is still informative: $27/45$ pairs are strongly linked (consistency $\geq 0.7$), $8/45$ are weak or unlinked ($\leq 0.3$), and the remaining $10/45$ are ambiguous. Representative high-consistency pairs include deer–dog (91%), bird–cat (91%), cat–deer (91%), cat–dog (82%), and automobile–truck (91%). Representative low-consistency pairs include frog–ship (18%), airplane–horse (18%), airplane–automobile (27%), cat–ship (27%), and deer–truck (27%). We use this summary, rather than the full 45-entry matrix, because the point is robustness of the signal across augmentation seeds and not a claim that each CIFAR-10 pair has an intrinsic binary linked/unlinked label.

### I.3. CNN architecture

To probe whether the bounded-width topological obstruction applies to CNNs on CIFAR-10, we use width-bounded CNNs whose every intermediate representation has flattened dimension $\leq 3072$ (matching the input). Each spatial-resolution stage quadruples channels while halving each spatial dimension, preserving the $C \cdot H \cdot W = 3072$ budget: $3 \times 32^2 \to 12 \times 16^2 \to 48 \times 8^2 \to 192 \times 4^2$. Depth variants L5/L8/L11 use 1/2/3 ConvBlocks per stage; ConvBlocks are $3 \times 3$ convs followed by activation (no batch normalization). When enabled, skip connections span each conv block: $x \mapsto x + \mathcal{G}(x)$. Training: Adam at $10^{-3}$, batch size 128, up to 100 epochs with patience-15 early stopping on the standard CIFAR-10 train/test split.

### I.4. 10-class confusion vs. linking

For each of $42$ trained models (7 activations $\times$ 3 depths $\times$ 2 skip settings) we compute the Spearman rank correlation between the linking-consistency scores and the symmetric confusion rate $\mathrm{conf}(i, j) = P(\hat{y} = j \mid y = i) + P(\hat{y} = i \mid y = j)$ across all 45 class pairs. The pattern is stable across L5/L8/L11 and across activation choice: mean Spearman $r \approx 0.47$ with $p < 0.003$ throughout. Representative L11/no-skip values are ReLU $r = 0.436$ ($p = 0.0028$), GELU $r = 0.501$ ($p = 0.0005$), and Mish $r = 0.497$ ($p = 0.0005$).

The same runs show a large separation in confusion rates. High-link pairs (consistency $\geq 0.7$) have about $3\times$ the confusion of low-link pairs (consistency $\leq 0.3$): ReLU gives 6.4% vs. 2.1%, GELU gives 6.0% vs. 1.8%, and Mish gives 5.9% vs. 2.1%. We report representative values rather than the full per-activation table because the rows are redundant; the relevant observation is that the correlation persists across architectures and activations.

### I.5. Distance-metric and projection controls

A natural concern is whether linking consistency simply measures inter-class distance. We compared linking consistency against two pixel-space distance metrics (mean Spearman $|r|$ across all 42 models): linking consistency reaches **0.479**, average pairwise distance 0.412, minimum-distance 0.356. Linking is therefore not reducible to a simple distance proxy; it captures additional signal about how class manifolds are intertwined rather than merely how far apart they sit on average.

We also repeated detection under 20 random 3D projections. In the same projection-control protocol, PCA detected links in $21/45$ class pairs, while random projections detected $31$–$43/45$ pairs depending on the projection. Per-pair random-projection frequency ranged from $11/20$ (frog–ship) to $20/20$ (deer–dog), preserving the same weak/strong ordering but with less discrimination than PCA consistency (55–100% vs. 18–91%). These random-projection frequencies vary projection subspaces, whereas the linking-consistency summary above varies augmentation seeds, so they should be read as complementary controls rather than the same estimator.

**Caveats.** The evidence above is suggestive, not causal. (i) Linking is detected in 3D PCA space, not the native 3072D pixel space, so some detected linking may be projection artifacts and some genuine high-dimensional linking will be invisible after PCA. (ii) The linked-pair gap on the binary task is modest (1.2%) and concentrated on the few hundred points that participate in detected cycles ($\sim 0.1\%$ of each class). (iii) The 10-class confusion correlation is robust but partially confounded by semantic similarity; the within-dataset comparison controls factors like resolution and collection methodology but cannot disentangle topology from semantic similarity entirely. Multi-projection aggregation and higher-dimensional topological data analysis are natural next steps for stronger conclusions.

