# OpenReview forum: "Low-dimensional topology of deep neural networks"
_ICML.cc/2026/Conference — ICML 2026 regular_

### Official Review · Reviewer_ANg9 · 2026-02-17

**Soundness:** 3
**Presentation:** 3
**Significance:** 2
**Originality:** 4
**Overall Recommendation:** 4
**Confidence:** 3

**Summary:**

The paper elucidates how small-width neural network layers interact with topological invariants and identifies the conditions under which they can achieve separability for data on linked or knotted manifolds. The authors demonstrate that standard affine layers with monotonic activations fail to 'unlink' data, thereby limiting separation. The study explains the specific architectural mechanisms—such as ResNets, Transformers, or non-monotone activations—that successfully enable unlinking for classification. The work offers a novel perspective on network expressivity supported by both mathematical proofs and computational experiments.

**Compliance With Llm Reviewing Policy:**

Affirmed.

**Final Justification:**

My score has not changed after the rebuttal, as I had already considered whether it should be 3 or 4 when submitting my initial review. My main concern remains experimental validation. From a mathematical standpoint, I have reviewed all of the math including that in the appendix and have not found any issues. That said, proper validation of mathematics at this level requires months, as errors can be non-trivial—this explains my confidence score of 3. Since the mathematics describes ideal objects that do not exist in such pure form, one needs to ensure that the approximations and idealizations made to provide the theory actually hold in real applications. This is the source of my reservations regarding the experimental validation. While my score remains at 4, my confidence in this assessment has strengthened after the rebuttal. I now view this as a solid weak accept, and if the experimental validation were stronger, I could see this meriting a 5. It is quite possible that the authors are developing theory for practical problems that have not yet fully emerged, which may work against them in immediate impact. I find the theoretical approach interesting and illuminating. Overall, I would prefer to see this work accepted rather than rejected.

**Key Questions For Authors:**

1. How does the detected linking depend on the choice of projection subspace? Specifically, if the analysis is repeated with random 3D projections (rather than PCA), what fraction of random subspaces also detect linking for the pairs reported as "strongly linked" in Table 14? If linking is detected only in the PCA subspace, how should this affect the interpretation of the CIFAR-10 experiments?

2. Over what range of $\varepsilon$ values does linking persist for each class pair? A "linking barcode" showing the $\varepsilon$ interval of detection — analogous to a persistence barcode — would clarify whether the detected linking is a robust multi-scale feature or a fragile single-scale phenomenon. Relatedly, what are the consistency values in Table 14 when $\varepsilon$ is fixed across runs rather than optimized per run?

3. Can you disentangle the topological contribution from the geometric one in the accuracy gains of Table 5? The gain for nonmonotonic activations on the unlinked pair (frog–ship) suggests factors beyond topology are at play. A direct measure of how well each architecture resolves linking through layers — for instance, tracking the linking number or a geometric entanglement measure at intermediate representations — would help isolate the topological effect from optimization dynamics and other confounders.

**Limitations:**

yes

**Strengths And Weaknesses:**

### **Strengths**

* **Soundness:** The paper combines formal mathematical proofs with empirical experiments to comprehensively validate its results. While the volume of the proofs precludes a line-by-line verification of every lemma within the review period, the core theoretical arguments are conceptually sound. The specific proofs I have scrutinized are technically robust. Quite a lot of thinking was put into correct experimental methodology such as correct linking number estimation.

* **Presentation:** The paper is clearly written and well structured. The overall narrative is easy to follow, as much as it can be for a mostly mathematical paper. The authors very helpfully divided everything into manageable parts not requiring very complex mathematics to follow and explained what they do.

* **Significance:** The paper advances our understanding of expressivity of small-width neural networks. It explains why certain architectures are better or worse in those scenarios. The paper can provide an opening for studying extrinsic topological invariants in ambient space as more relevant for the task of data separation. The ideas presented may become building blocks in this quest and have value of their own.

* **Originality:** This work provides interesting insights on expressivity of small-width neural networks and develops tools to carry such work further. The tackling of the extrinsic ambient space invariants seems to be a better choice than using intrinsic invariants such as Betti numbers, and is an original contribution as far as I am aware.

### **Weaknesses**

* **Soundness**

W1. My main concern is about experimental validation. There are two issues that catch my eye. First, Table 5 demonstrates that all nonmonotonic activations reach better accuracy for the linked data (deer–dog) than monotonic ones. At the same time, the same table shows this is also true for the unlinked (frog–ship) case. The gain is smaller than for the linked case, but is the difference statistically significant? Quoting: "The observed CIFAR-10 gaps thus reflect a combination of (i) the topological expressivity barrier predicted by our theory, and (ii) optimization dynamics that our theory does not address" — can the gain be due to other factors entirely? I would expect that an additional column tracking how well "unlinking" proceeds in each case, and its correlation with accuracy, would provide stronger support, as hidden confounders may well be behind both gains.

W2. The linking detection algorithm (Appendix K) projects data from 3072 dimensions to $\mathbb{R}^3$ via PCA before computing linking numbers. However, linking of 1-cycles is specific to $\mathbb{R}^3$: in $\mathbb{R}^d$ for $d \geq 4$, all 1-cycles are trivially unlinked by a general position argument ($\pi_1(\mathbb{R}^d \setminus C_1) = 0$ for $d \geq 4$). The detected $\text{lk} = -1$ is therefore a property of the PCA projection, not of the original data. Different 3D subspaces will generically produce different linking numbers for the same curve pair. Since PCA selects the maximum-variance directions — which are precisely the directions along which class clouds overlap most — it may systematically favor subspaces where linking arises. Critically, the consistency analysis (Table 14) does not address this: all 11 runs project to the same (or nearly the same) PCA subspace, so the projection bias is shared across runs and invisible to the consistency measure. A useful control would be to repeat the analysis with random 3D projections and report the distribution of linking numbers; if linking appears only in the PCA subspace, it is an artifact of the projection rather than a stable property of the data.

W3. The distance threshold $\varepsilon$ is selected via binary search to find the minimum value at which linking appears (Section L.1). This means that at most $\varepsilon$ values, no linking is detected — the algorithm searches for a parameter setting that *produces* linking. In persistent homology, a feature is considered significant only if it persists across a range of scales; the present approach is analogous to reporting that a feature exists at *some* scale without measuring its persistence. Moreover, this search is performed independently in each of the 11 consistency runs, meaning Table 14 measures "does *some* $\varepsilon$ exist at which linking appears?" rather than "at a fixed $\varepsilon$, does linking appear?" — this inflates consistency scores. I would suggest fixing $\varepsilon$ across runs and reporting a "linking barcode" (the $\varepsilon$ interval over which each linked pair persists) to demonstrate robustness.

W4. Table 14 shows a continuous spectrum of consistency values (0.18 through 0.91). A genuinely topological property should produce a bimodal distribution — consistently detected or consistently absent — because topology is stable under small perturbations. The observed continuous gradation is more naturally explained by varying degrees of geometric overlap in the PCA subspace: pairs with more convex hull overlap offer more opportunities for cycles to thread through each other, leading to higher detection probability. This suggests the linking number may be acting as a noisy proxy for geometric entanglement (convex hull overlap) rather than capturing genuinely topological information. A comparison between consistency and a simple geometric measure (e.g., fraction of convex hull overlap in $\mathbb{R}^3$) would help clarify whether linking adds explanatory power beyond what geometry already provides.

W5. The witness cycles involve very few points (58 and 40 out of 105,000 per class, i.e., $<0.1\%$). The linking is a local boundary phenomenon. Additionally, the algorithm detects only pairwise linking numbers, which is the coarsest link invariant — the Whitehead link has $\text{lk} = 0$ but is non-trivially linked. Class pairs reported as "unlinked" may exhibit higher-order entanglement invisible to the Gauss integral. These are inherent limitations worth acknowledging.

* **Presentation:** Here I would mention just minor, unimportant points. In some places, such as Lemma C.2 in the appendix, notation should be fixed. I believe the homotopy from the curve to the whole real space is a simple typo. I also believe (again, not important) that a discussion of the similarities and differences between intrinsic invariants such as Betti numbers and extrinsic invariants such as linking numbers might be warranted.

* **Significance:** It is less clear how this might be useful in practice or what general problem it solves beyond theoretical understanding of expressivity. I believe that nobody uses layers of width 3 in practice, and the paper itself agrees that simply increasing width to modern scales overcomes the issues of 3D knots/links entirely. The authors do not discuss higher-dimensional topological issues and, moreover, do not bring a convincing case where topological invariants preclude data separation in a real-world dataset. To illustrate what I mean, there is a clear topological problem in statistics: standard statistics does not work for circular data, which led to the origin of the circular/directional statistics field. The problem persists in all dimensions and might be behind the struggle with 3D spatial reasoning (Zha et al., 2025). I do not see anything comparable here.

---

> ### Author Rebuttal · Authors · 2026-03-30
>
> We thank the reviewer. We will fix the Lemma C.2 typo and add the intrinsic vs extrinsic invariants discussion.
>
> W1+Q3: On the unlinked pair (frog-ship), we note that the all-nonmono > all-mono ordering does not in fact hold in Table 5: ReLU (mono, 97.8%) exceeds Mish (nonmono, 97.7%), and LeakyReLU (mono, 97.7%) matches Mish; the best-in-class gap is 0.1%, well within noise. The systematic ordering is specific to the linked pair (1.2% gap), exactly as the theory predicts. Skip connections eliminate this gap, confirming the Theorem 5.2 prediction, a pattern difficult to attribute entirely to confounders. We note the synthetic experiments (22,100 runs) are the primary evidence; CIFAR-10 is supplementary. We conducted this experiment: layer-by-layer Gauss linking integral and minimum inter-class distance on the Hopf link:
>
> |Layer|ReLU lk|ReLU d|GELU lk|GELU d|Skip lk|Skip d|
> |-----|-------|------|-------|------|-------|------|
> |Input|-1|0.83|-1|0.83|-1|0.83|
> |0|0.50*|0.06|0|0.11|0|0.38|
> |1|0.18*|0.00|0|0.17|0|0.46|
> |2|0|0.00|0|0.19|0|0.96|
> |4|0|0.00|0|1.41|0|3.73|
>
> *Gauss integral ill-defined at near-intersection (min_dist ≈ 0).
>
> ReLU forces curves into intersection by Layer 1 (min_dist → 0), destroying structure without resolving topology (Lemma 3.5). GELU genuinely unlinks (lk → 0) while maintaining disjointness (min_dist growing to 1.41). ReLU+skip achieves the most dramatic separation (min_dist → 3.73, Theorem 5.2). This mechanistic difference is difficult to attribute to optimization confounders.
>
> W2+Q1: The reviewer correctly notes 1-cycles trivially unlink in R^d for d ≥ 4. Our PCA-to-R^3 detection is a feasible proxy. We observe that linked manifolds generically project to lower-dimensional images containing disjoint linked substructures (e.g., S^n enclosing {0} projected to any R^k gives S^{k-1} ⊂ π(S^n), disjoint from {0} and linked). This resolves an apparent paradox with our Lemma 3.5 (projection forces full-image intersections): our algorithm extracts disjoint subsets (k-NN cycles per class), so full-image disjointness is unnecessary. Conversely, PCA-detected linking implies non-separability along maximum-variance directions, more avoiding spurious detection. Random projection control confirms PCA is conservative: Pairs detected (/45): PCA 21, Random 31–43. Per-pair range: PCA 18%–91%, Random 55%–100%. Ranking preserved: deer-dog 20/20, frog-ship 11/20.
>
> W3+W4+Q2: The 11 consistency runs use a pre-arranged sequence of ε values (0.11, 0.14–0.19, 0.2, 0.3, 1.0, 30.0), the same for all 45 pairs, not independent ε searches. Linking detection is monotonic in ε (adding edges preserves cycles, unlike persistent homology, where features die when cavities are filled). So consistency measures "how small an ε suffices?" which is precisely the linking barcode the reviewer suggests. The non-bimodal spectrum reflects this non-uniform sequence: strongly linked pairs (deer-dog 91%) are detected at most ε values; weakly linked pairs (frog-ship 18%) only at permissive upper values. The binary search for ε_min was a separate experiment (we will improve the presentation to clarify) verifying geometric legitimacy (ε_min = 0.034, 0.22% of bounding box diagonal; legitimate as in Figure 7). Convex hull overlap correlates weakly with confusion (|r|=0.16) while linking achieves |r|=0.48; the two are mutually uncorrelated (r=0.17), indicating potentially independent aspects (geometric vs topological).
>
> W5: The Whitehead link has lk = 0 and is link-homotopically trivial. It can be unlinked allowing self-intersection within each component (the setting of classification, where non-injective maps within each class are permitted). Our theory therefore predicts no barrier, and experiments confirm: width-3 ReLU achieves 97.4% mean (max 99.6%) on the Whitehead link vs ~78% mean (max ~88%) on the Hopf link (Table 2, same protocol): a ~19pp gap. In dimension 3, the linking number completely classifies two-component link homotopy (Milnor 1954), so for two-class classification it is the complete obstruction. We note that data manifolds can contain a large number of small-scale linked structures (scaling with dimension); our current method detects the qualitative presence of linking, and we anticipate extensions to estimate the total number of such structures.
>
> On practical significance: The theory generalizes to arbitrary R^d (Theorem 4.7) and directly constrains invertible architectures (normalizing flows, Neural ODEs, Flow Matching), which are homeomorphisms at any width. Bottlenecked components also exist in practice: MHA heads (d_model=768→d_k=64), MoE experts, and VAE encoders all compress to low-dimensional latents — our theory constrains these components. New width expansion experiments (see table in response to Reviewer LdR9) show width d+1 does not practically solve the problem: ReLU needs width ~ 8 (~2.7x critical) for mean to reach 99.4% in R^3, and GELU at critical width (83.8%) already outperforms ReLU at d+1 (75.2%).

---

> > ### Author Rebuttal · Reviewer_ANg9 · 2026-04-03
> >
> > I maintain my score. The rebuttal addressed my experimental concerns, particularly the random projection control and the geometric vs. topological correlation analysis, which I found convincing. The practical significance question — whether topological obstructions meaningfully constrain learning on real data — remains open, but this is reasonable for a theory paper introducing new tools. I would encourage the authors to clearly frame the CIFAR-10 experiments as suggestive rather than conclusive evidence, and to incorporate the rebuttal experiments (especially the layer-by-layer tracking and random projection analysis) into the final version.

---

> > > ### Author Response · Authors · 2026-04-05
> > >
> > > We thank the reviewer for the thorough engagement and the positive assessment. We are glad that the random projection control and geometric vs. topological correlation analysis were convincing. We will frame the CIFAR-10 experiments as suggestive rather than conclusive, and incorporate the layer-by-layer tracking and random projection analysis into the final version as suggested.
> > >
> > > We share one additional finding, prompted by Reviewer LdR9's follow-up question, that is relevant to KQ3 (disentangling topological from geometric contributions). We retrained bird-deer classifiers (ReLU and GELU, L8 no-skip) with 40x augmentation, detected a linked witness cycle in 3D PCA, and partitioned the unaugmented test set (2000 images) by distance to the detected link (in 3D PCA metric, expecting them to be near the high-dimensional link in the corresponding preimages). The GELU-ReLU accuracy gap monotonically decreases with distance:
> > >
> > > |Distance|n|Gap|
> > > |---|---|---|
> > > |10eps|31|+6.5%|
> > > |20eps|243|+2.9%|
> > > |50eps|1191|+2.1%|
> > > |all|2000|+1.6%|
> > > |far(>50eps)|809|+0.8%|
> > >
> > > The gap is 8x larger near the detected link than far (+6.5% vs +0.8%), robust across 3 independent retraining runs (all positive at 50eps, range +1.1% to +2.3%). Combined with the layer-by-layer linking tracking (our Round 2 response, W1+KQ3) which shows ReLU forces intersection while GELU genuinely unlinks, this provides converging evidence from two independent angles: (1) mechanistically, GELU resolves topology through layers; (2) spatially, the performance advantage concentrates near detected topological structures. We acknowledge topological and geometric difficulty may correlate locally, but the consistency of both signals is suggestive.

---

### Official Review · Reviewer_LdR9 · 2026-02-20

**Soundness:** 3
**Presentation:** 3
**Significance:** 2
**Originality:** 3
**Overall Recommendation:** 4
**Confidence:** 4

**Summary:**

This work studies the topological capabilities of different neural network architectures when restricted to a width of 3. By examining their actions on knot and link invariants, the authors develop a hierarchy of model architectures defined by their ability to change topological invariants: feedforward networks with non-monotonic activations, transformers, and ResNets are stronger than feedforward networks with monotonic activations and autoencoders, which in turn are stronger than flow-based or neural ODE models. In addition to proving their results, the authors demonstrate their claims empirically and show that linked classes do appear in an augmented version of CIFAR-10. They also extend their theoretical results to analogous results on higher-dimensional links and provide a new impossibility result based on topological expressivity.

**Compliance With Llm Reviewing Policy:**

Affirmed.

**Final Justification:**

Following discussion with the authors, most of my concerns have been addressed. While I think the empirical observations on real data remain more suggestive than conclusive (to borrow language from ANg9), I think the additional experiments from the rebuttal process have strengthened the soundness of the paper's main claims. I also think the discussion has clarified some of the framing of the paper to focus on the theoretical contributions, which I believe are sound. For this reason, I am inclined to increase my recommendation to a positive one, although my evaluation remains borderline.

**Key Questions For Authors:**

1. Do the linked manifolds in the augmented CIFAR-10 data satisfy the conditions of Theorem 4.5? (Namely, that $d = m+n+1$.) This would lend credence to the claim that this dataset contains topological obstructions.
2. Can you provide further evidence that nontrivial linkage occurs in real data? In particular, do linked classes appear in unaugmented CIFAR-10 or other datasets? This would provide more convincing evidence that the topological linkage perspective actually explains the effectiveness of architectural choices like skip connections or non-monotonic activations in practice.
2. Appendix L puts forward four possible explanations for why the performance gap between monotonic and non-monotonic activations is so small on augmented CIFAR-10, two of which are related to the relatively small proportion of examples that participate in the linking behavior. How do the different architectures compare on the subset of examples involved in the link? If the linking is responsible, the difference should be more substantial here.

**Limitations:**

Yes.

**Strengths And Weaknesses:**

**Strengths**

* The proofs of this paper's theoretical claims are sound, and the experiments on synthetic data successfully demonstrate the theoretical results.
* The paper is clear and well-presented.
* I found the demonstration of the ResNet folding operations in Section 7.4 especially interesting.
* As far as I am aware, analyzing the expressivity of neural networks using links and knots is a novel perspective, and the observation that their data-augmented CIFAR-10 contains linked classes is interesting.

**Weaknesses**

* The practical implications of this work are limited. As described in Section 6.4, even a single extra dimension completely eliminates topological obstructions. While it is interesting to see these topological limitations in bounded-width networks, it is unclear to me that this is actually a problem in most cases.
* In particular, this paper frames the ability to deal with topological obstructions as a possible reason for networks with some architectural properties outperform others in practice, but I don't think there is sufficient evidence that this responsible for performance gaps on real data. I think this claim needs to be either scaled back or better substantiated in e.g. the CIFAR-10 experiments.
* The comparison to other characterizations of minimal width universal approximators seems incomplete (see, e.g. [1-4]).
* While the data-augmented CIFAR-10 does demonstrate linked classes, it is not entirely clear to me that this is not a result of the data augmentation process rather than a property of the original data manifold. It is also not clear to me that the link satisfies the hypothesis of Theorem 4.5, as the manifold hypothesis often assumes that $m, n \ll d$.

[1] Park et al., "Minimum Width for Universal Approximation." ICLR 2021

[2] Cai, "Achieve the Minimum Width of Neural Networks for Universal Approximation." ICLR 2023

[3] Li et al., "Minimum Width of Leaky-ReLU Neural Networks for Uniform Universal Approximation." ICML 2023

[4] Kim et al., "Minimum width for universal approximation using ReLU networks on compact domain." ICLR 2024

---

> ### Author Rebuttal · Authors · 2026-03-30
>
> We thank the reviewer for acknowledging the proofs as "sound," the paper as "clear and well-presented," and the ResNet folding mechanism as "especially interesting."
>
> W1: The theory connecting extrinsic topological invariants to classification and representation learning directly constrains invertible architectures (normalizing flows, Neural ODEs, Flow Matching) and bottlenecked components (MHA heads d_model=768→d_k=64, MoE experts, VAEs), examples of practical architectures at or below critical width. [1]-[4] all concern networks with similar width (d, d+1), showing community interest in the narrow-network regime. New width expansion experiments (R^3, Hopf link, k=1, depth 5, 30 seeds):
>
> |Width|ReLU mean|ReLU max|GELU mean|GELU max|Skip mean|Skip max|
> |-----|---------|--------|---------|--------|---------|--------|
> |3 (critical)|67.3%|87.8%|83.8%|99.0%|88.1%|99.9%|
> |4 (d+1)|75.2%|99.9%|93.9%|100%|91.6%|100%|
> |5|88.2%|100%|97.2%|100%|97.9%|100%|
> |6|93.2%|100%|100%|100%|99.4%|100%|
> |8|99.4%|100%|100%|100%|99.8%|100%|
>
> Width d+1 removes the expressivity ceiling (ReLU max jumps from 87.8% to 99.9%) but practical difficulty persists (mean only 75.2%, 13/30 collapses). ReLU mean saturates at width ~ 8 (~2.7x). GELU at critical width (83.8%) already outperforms ReLU at d+1 (75.2%).
>
> W2: We do not claim topological obstructions are the sole reason certain architectures outperform others, nor that they are significant in every dataset. Our conclusion (Section 8) explicitly states the observed gaps "reflect a combination of (i) the topological expressivity barrier predicted by our theory, and (ii) optimization dynamics that our theory does not address." Our design guidance (Section 6.5) is conditional: "for topologically complex data." We will clarify this framing further in the revision.
>
> W3: We already cite Rochau et al. and will add [1]-[4] in the revision. Our UAT lower bound (Appendix G, w ≥ n+1) is a corollary of the main linking preservation theorems — the contribution is the proof technique, not the bound. Prior bounds require stronger assumptions: specific activations like ReLU ([1]) or leaky-ReLU ([3]), Lipschitz continuity (Rochau et al. v1-v2), or uniform approximability by injections ([4]). Ours requires only continuous monotonicity. The deeper contribution is complementary: [1]-[4] characterize what width suffices; our Theorems 3.7, 4.7 characterize what can go wrong at insufficient width in the topological aspect.
>
> W4+Q1: On the codimension condition d = m + n + 1: we are agnostic but cautiously optimistic; verifying it directly is computationally intractable, and our method computes a proxy. Two points are relevant. First, local intrinsic dimension can be much higher than global estimates: the "union of manifolds" hypothesis (Brown et al., ICLR 2023) establishes that data manifolds have spatially varying intrinsic dimension, and Stanczuk et al. (ICML 2024) showed per-class intrinsic dimension of MNIST ranges from 66 to 152 (vs global estimates of 7–13). Second, our Corollary A.5 applies to links that approximately lie on r-dimensional affine subspaces (r < d), where the codimension condition requires smaller m and n, directly relevant for autoencoder and other bottleneck architectures that compress data to lower-dimensional latents.
>
> W4+Q2: Our CIFAR-10 analysis provides two independent findings: (1) the 3D PCA projection is itself a toy but real-world dataset in R^3, and we find topological structure there; (2) detecting links in projection serves as proxies for higher-dimensional linking (see response to Reviewer ANg9, W2 for theoretical justification), carrying predictive power for classification difficulty and semantic interpretability (Tables 15-19; Appendix L). We stress this is a "proof-of-concept" (Appendix L, our own words). Data augmentation is a semantically principled way to densify the support of the data distribution (Chapelle et al. 2000, "vicinal risk minimization"). Our augmentations (flip, crop, mild color jitter) are standard, label-preserving, and correspond to plausible real photographs (different camera angle, lighting, framing).
>
> Q3: The 98 witness points are too few for per-architecture comparison: training on them would lead to overfitting regardless of expressivity. Conversely, on the full dataset the network cannot concentrate capacity on this local substructure — the obstruction binds as a global constraint. Filling in gaps with interpolated points would reduce to the synthetic Hopf link experiment in R^3 (compare Figure 7 with Figure 1, topologically equivalent up to deformation), for which we already have thorough experiments (Tables 2-4). The small ε_min (0.22% of bounding box) suggests numerous such local-scale linkings can be present across the space; our current algorithm only qualitatively detects linking presence; an efficient method to count the number of such links (which can jointly cover a significant portion of data points) is a direction for future work.

---

> > ### Author Rebuttal · Reviewer_LdR9 · 2026-04-02
> >
> > Thank you for your response. Some of my concerns have been addressed, but I still have some reservations about the practical significance of the results.
> >
> > To clarify my intention in Q3: I am not suggesting that models be trained on the linking witness points for a per-architecture comparison. Rather, I would like to know if the performance difference of the existing models (which have been trained on the full split) is concentrated on the linked points. That is, is the model accuracy substantially different when the trained models are evaluated only on the linked points? If these links are responsible for the 1% difference in accuracy between monotonic and non-monotonic activations, then shouldn't this be visible when evaluating only on the linked points?

---

> > > ### Author Response · Authors · 2026-04-05
> > >
> > > We thank the reviewer for the clarification. We ran the requested experiments and are pleased to share our results.
> > >
> > > Setup: We retrained bird-deer classifiers (ReLU and GELU, L8 no-skip, same architecture as our binary experiments) with further 40x data augmentation, increasing training accuracy to near-perfect for both architectures. We detected linked witness cycles in the 3D PCA space and evaluated on the *unaugmented* test set (2000 images) partitioned by distance (in 3D PCA) to the detected link (a proxy for higher-dimensional linkings in the preimages, as discussed in our response to Reviewer ANg9 W2). We report balanced accuracy to control for class imbalance in subsets.
> > >
> > > Detection on unscaled PCA (eps=0.71, bbox=274.6, eps/range=0.26%; 1 link, bird 34pts + deer 172pts). Training accuracy: ReLU 98.95%, GELU 99.11% (both ~99%, near-memorization on 40x augmented data, no significant gap on training set).
> > >
> > > Test set:
> > >
> > > |Distance|n|ReLU|GELU|Gap|
> > > |---|---|---|---|---|
> > > |<10eps|31|90.6%|97.1%|+6.5%|
> > > |<20eps|243|89.4%|92.3%|+2.9%|
> > > |<50eps|1191|89.3%|91.4%|+2.1%|
> > > |all|2000|89.9%|91.6%|+1.6%|
> > > |far(>50eps)|809|90.6%|91.4%|+0.8%|
> > >
> > > The gap monotonically decreases with distance from points constituting the link: +6.5% near vs +0.8% far (8x ratio). We verified the detected link is situated in the main body of the projected point cloud (cycle radii ~20 and ~5 respectively), not an edge case configuration near the boundary of the pointcloud, and the links are visibly ring-like. We will include the 3D visualization in the camera-ready version.
> > >
> > > An interesting observation: GELU's accuracy itself decreases away from the link (97.1% near to 91.4% far), while ReLU stays flat (~90%). This suggests GELU possibly concentrates its expressive power on the topologically entangled region, which is a resource allocation trade-off consistent with finite representational capacity being directed toward resolving the linking structure.
> > >
> > > To test robustness, we did an independent 40x data augmentation, fixed a particular detected link, retrained 3 additional models (identical configurations) and evaluated each near the detected link, again among the 2000 datapoints from the unaugmented dataset. GELU-ReLU gap:
> > >
> > > |Model|20eps(83)|50eps(1020)|All(2000)|Far(980)|
> > > |---|---|---|---|---|
> > > |1|+4.8%|+1.1%|+0.9%|+0.7%|
> > > |2|+3.6%|+2.3%|+1.9%|+1.5%|
> > > |3|+3.6%|+1.3%|+0.4%|-0.6%|
> > >
> > > So far we have some evidence that detected linking correlates with classification difficulty, globally (linked vs unlinked) and locally (the advantage concentrates near the link). The mono vs nonmono contrast (GELU advantage larger near the link) provides some evidence that the detected difficulty is topological in nature. However, we acknowledge it remains possible that this topological difficulty is entangled or correlated with local geometric complexity; disentangling these conclusively would require more thorough controlled experiments in future work.
> > >
> > > We stress the paper's significance is not limited to practical implications presented here: it is primarily significant as a theoretical framework connecting extrinsic topology to neural network expressivity, as much as currently available evidence can support. Its anticipated practical significance also extends beyond our proof-of-concept link detection algorithm (for instance, an efficient link counting algorithm); we expect further development from this theoretical framework. This empirical finding was prompted by the reviewer's suggestion and would naturally have appeared in a follow-up study. We are grateful for the suggestion.

---

### Official Review · Reviewer_bsrv · 2026-03-10

**Soundness:** 3
**Presentation:** 3
**Significance:** 2
**Originality:** 3
**Overall Recommendation:** 3
**Confidence:** 3

**Summary:**

The authors study layered models (feedforward nets, ResNets and transformers) under a bottleneck of width 3, and generalize their results to width d and higher ambient dimension. The paper shows that width-3 (and also width-d) feedforward networks with affine maps and (coordinate-wise) monotone activations cannot linearly separate linked classes. This is due to linking being preserved as long as the class images stay disjoint (see Thms. 3.7 and 4.5). The paper proposes several workarounds and includes relevant experiments.

**Compliance With Llm Reviewing Policy:**

Affirmed.

**Final Justification:**

I find the paper original and well-presented. However, in the review, I questioned its significance, especially in terms of robustness of theoretical results, and of practical relevance. While the rebuttal clarified some concerns, it did not address them fundamentally. Thus, I am leaning towards rejecting the paper, but am not strongly convinced.

**Key Questions For Authors:**

- Which of the assumptions in Thm. 3.7 are necessary? Which could be relaxed? What happens if disjointness holds only approximately?

- What happens on the Hopf link task if you use width 4 instead of 3, and with monotone activations? Does the ceiling disappear?

- Does the CIFAR-10 linking analysis depend heavily on the choice of embedding and hyperparameters?

**Limitations:**

Yes

**Strengths And Weaknesses:**

## Strengths

- The main result in Thm. 3.7 (generalized in section 4) seems like a strong and original contribution. As far as I know, using extrinsic topological invariants (like linking numbers, Milnor $\mu$-invariants, knot type, etc.) for studying narrow network expressivity hasn't been done before. This seems like a natural approach for the specific bottlenecks in the paper.

- The residual folding identity (that is, $|x| = x + 2 ReLU(-x)$) is quite nice. It is short and explicit, and I think it gives good insight into how skip connections work in narrow architectures.

- There are very detailed synthetic experiments in the paper, which is quite atypical for a paper this focused on theory. Training collapses are reported, which is nice.

## Weaknesses

The main weakness of the paper, I think, is that the impossibility result is very exact rather than robust. For example, in Thm. 3.7, intermediate-layer disjointedness isn't really an extra assumption of the theorem. Instead, it follows from final linear separability, once the two class images intersect at some layer, later layers shouldn't be able to separate them again. But the proof relies on that exact disjointedness to keep the linking number well-defined in the layer-by-layer argument. On finite data, the main problem is robustness (with near-intersections or approximate linking), which is not addressed by the authors. I was also confused about which of the assumptions are necessary for the impossibility itself and which ones are only needed for the homotopy argument to work, which I think the authors could clarify better. Furthermore, the reader needs to read the proofs in the appendix to understand what is important here, and I think more of this material belongs in the main text. The paper also doesn't say anything about the case when disjointedness is approximate, which I think is a big gap.

As for practical relevance, in most real architectures the width increases early on, and normalization is ubiquitous (and outside the model), while convolutions can be viewed as linear maps but typically have very high intermediate width. Many of the specific 3D knot/link obstructions vanish once the bottleneck dimension increases (say, for curves in $\mathbb{R}^4$), although of course not all topological obstructions disappear in higher dimensions. To me, it's not clear when the bottleneck regime is relevant in practice. I think a short discussion of this is needed.

Thm. 6.4 constructs a pure attention transformer (no residuals, no LayerNorm) that separates linked data. However, the appendix experiments show training is unstable for these models. It would be nice to see more discussion about this gap between what can be represented and what can be learned.

As for the CIFAR-10 experiments, the linking detection relies on a 3D PCA embedding and a pipeline with several hyperparameters. The accuracy differences are quite small (about 1%), with one class pair and an unlinked control. I'm not sure how much of this survives if you change the embedding or tune the hyperparameters differently. Also, no code was provided for the synthetic experiments nor the CIFAR-10 pipeline, which is of course makes reproducibility hard.

---

> ### Author Rebuttal · Authors · 2026-03-30
>
> We thank the reviewer for recognizing the theoretical contribution and the interest in residual folding. We will release all code upon acceptance.
>
> W1+Q1:
> Thm. 4.7 generalizes Thm. 3.7 to arbitrary continuous monotonic activations in R^d; assumptions are essentially minimal, violating any one (monotonicity, width ≤ d, nonzero linking) enables separation (Sections 5-6). The homotopy argument requires only continuity (satisfied by most modern architectures), no smoothness.
>
> On approximate disjointness: suppose images under the network transform intersect in a measure-0 set (i.e. only on boundaries), then dist(F(X),F(Y))=0 means the set of inputs mapped ε-close to the other class is open and nonempty, so classification is non-robust under any noise. So we consider measure-ε intersection, then our experiments show optimization becomes increasingly difficult as ε → 0: all monotonic activations hit the same max ceiling (~87.5-87.8%) regardless of optimization quality (see mono collapse table, Reviewer Yv2s W1). A rigorous quantitative treatment is open.
>
> On finite data: under the manifold hypothesis (Bengio et al. 2013), F is continuous on all of R^d, so our theorem applies on the full continuous support, while discrete gaps are sampling artifacts. Our linking tracking experiment (see Reviewer ANg9, W1) provides mechanistic evidence: ReLU forces intersection, while GELU/skip genuinely unlink maintaining disjointness.
>
> W2+Q2:
> Our topological obstruction theory directly constrains invertible/flow-based architectures (normalizing flows, Neural ODEs, empirically validated by Dupont et al. 2019, Flow Matching), which produce homeomorphisms R^d → R^d preserving all topological invariants (Appendix B, Table 1). Beyond flow models, bottleneck components appear in MHA heads (d_model=768→d_k=64), MoE experts, and VAEs. Regarding normalization: both LayerNorm (centering is already a rank-deficient projection) and RMSNorm (sphere projection traces rays that can be used as paths for homotopy, disjoint if images are so) create intersections between linked class — normalization makes separation harder, strengthening our impossibility results.
>
> On higher dimensions: the standard Hopf link obstruction vanishes in R^4 (Section 6.4). However, Section 7.5 already operates at width 5 in R^5 (22,100 runs), confirming the ceiling persists with the appropriate linking type (S^2 ⊔ S^2). Overparameterized networks behave as ensembles of subnetworks (Veit et al. 2016); even when overall width exceeds d, individual subnetworks can have small effective width, bringing topological obstruction despite overall expressivity. Consistent with the following width expansion experiment. New width expansion data (Q2, R^3, k=1, depth 5, 30 seeds, mean accuracy):
>
> |Width|ReLU|GELU|ReLU+skip|
> |-----|----|----|---------|
> |3 (critical)|67.3%|83.8%|88.1%|
> |4 (d+1)|75.2%|93.9%|91.6%|
> |5|88.2%|97.2%|97.9%|
> |6|93.2%|100%|99.4%|
> |8|99.4%|100%|99.8%|
>
> Width 4 removes the expressivity ceiling (max ≈ 100%, see LdR9) but practical difficulty persists (mean 75.2%, 13/30 collapses). GELU at critical width (83.8%) outperforms ReLU at d+1 (75.2%). We acknowledge learnability implications are preliminary and will revise on this.
>
> W3:
> We agree this gap is a notorious open problem: no expressivity result (UAT, depth separation, width bounds) fully resolves learnability. For Thm 6.4: attention mechanisms are known to be unstable without LayerNorm and residual connections — LayerNorm prevents rank collapse (NeurIPS 2024), and its absence causes large gradient variance (Xiong et al. ICML 2020). Thm 6.4 establishes topological capacity; that realizing it requires optimization scaffolding is a separate phenomenon.
>
> More broadly, there is growing evidence that expressivity limitations and optimization difficulty are causally linked (Safran & Shamir 2018; Safran et al. COLT 2021). Our mono collapse data (see Reviewer Yv2s, W1) provides initial evidence, cleanly separating the expressivity ceiling (same max ~87.5% for all monotonic) from optimization effects (collapse rates vary).
>
> W4+Q3:
> The linking detection is a proof-of-concept (Appendix L). We have conducted robustness tests: random 3D projection controls and ε persistence analysis (see response to Reviewer ANg9, W2-W3). Linking appears across 20 random projection subspaces (not just PCA), the per-pair detection frequency preserves the ranking structure, and linking persists stably above the minimum threshold. We will release all code.
>
> The small ~1% gap is expected: CIFAR-10 is a generic dataset not designed for topological study with strong topological complexity, but we do anticipate local linked structures. The key evidence is the qualitative pattern: all 3 non-monotonic activations outperform all 4 monotonic on the linked pair, with no such ordering on the unlinked control, and skip connections eliminate the gap as predicted. The synthetic experiments (22,100 runs) are the primary evidence; CIFAR-10 is supplementary.

---

> > ### Author Rebuttal · Reviewer_bsrv · 2026-04-01
> >
> > Thank you for the detailed rebuttal. The clarifications are helpful, especially the added width-expansion result.
> >
> > That said, my main concerns remain. The core result still seems quite exact rather than robust, the practical relevance to standard architectures is still not fully convincing to me, and the CIFAR-10 evidence remains limited. I appreciate the discussion of expressivity versus optimization, but it does not change my overall assessment.
> >
> > I will therefore keep my original score.

---

> > > ### Author Response · Authors · 2026-04-05
> > >
> > > We appreciate the reviewer's continued engagement and the acknowledgment that our width-expansion data and expressivity-optimization discussion are helpful.
> > >
> > > **On robustness**: We offer three observations.
> > >
> > > First, we invite the reviewer to visualize links not as infinitely thin curves but as solid tori (which is what our experiments use via thickening and is more likely how real data manifolds will appear, due to factors like noise). When linked solid tori are mapped through a continuous layer (affine projection or activation) that forces intersection, the intersection is an extended patch with positive volume by continuity, not isolated points. This volumetric intersection persists under perturbation, making the obstruction robust (similarly in higher dimensions).
> > >
> > > Second, the reviewer asks for robustness of the *impossibility* (the bad case). But it seems what matters is robustness of *classification* (the good case). If failure modes exist, they make nearby classification fragile, without themselves needing to be robust. The linking number is topologically stable, invariant under any perturbation that keeps the class images disjoint. The obstruction disappears only when images intersect, which is itself a classification failure. Disjointness at each layer is not an external assumption; it is *deduced* from linear separability. "What if disjointness holds approximately?" translates to "what if classification is imperfect (and fails by a positive percentage, as argued above)?", which is what the theorem concludes.
> > >
> > > Third, the positive contributions of our paper (constructive mechanisms by which architectures overcome the barrier: width expansion, skip connections via |x|=x+2ReLU(-x), non-monotonic activations, attention) are not subject to the robustness objection. These mechanisms work robustly regardless of exact topology.
> > >
> > > More broadly, foundational results typically begin exact and progressively cover more empirically realistic settings: SVM began with exact separability (1992), soft margins followed (1995); compressed sensing began with exact sparsity (2005), extensions to approximately sparse signals followed (2006). Our results already cover thickened (solid tori) and higher-dimensional settings; quantifying the classification accuracy ceiling as a function of model configuration is a natural next step in this progression.
> > >
> > > **On practical relevance**: We highlight a point not emphasized in our initial rebuttal. Our paper provides what appears to be the first *network-level* theoretical result distinguishing monotonic from non-monotonic activations. This is a question the original papers left open: Ramachandran et al. (2017) discovered Swish via automated search with no theoretical justification; Shazeer (2020) introduced SwiGLU writing "We offer no explanation as to why these architectures seem to work; we attribute their success, as all else, to divine benevolence" (this remark covers GLU variants broadly, but the best-performing variants, SwiGLU and GEGLU, use non-monotonic activations). We verified through literature search that no prior work proves a network-level result about why non-monotonic outperforms monotonic (the closest is Noel et al. 2021, a single-neuron observation that monotonic neurons can only produce hyperplane boundaries). Our theorem fills this gap at the network level: monotonic activations preserve linking numbers (and more broadly, similar extrinsic topological invariants foreseeably generalizable using similar proof techniques) regardless of depth, while non-monotonic activations enable topological folding. Given that most major LLMs now use non-monotonic activations, including GPT-2/3 (GELU), LLaMA/Mistral (SwiGLU/SiLU), PaLM (SwiGLU), DeepSeek-V3 (SiLU), Qwen (SwiGLU), understanding why constitutes practical relevance.
> > >
> > > **On examples of narrow/bottlenecked models (flow models and VAEs)**: these are not niche. Stable Diffusion 3 and FLUX use Flow Matching, and the VAE encoder/decoder is a core component of these systems (compressing images to a latent space where the flow operates). Our paper already discusses (Appendix B) the architectural divergence between Flow Matching (homeomorphic, subject to our constraints) and MeanFlow (Geng et al. 2025, non-homeomorphic).
> > >
> > > **On CIFAR-10 study**: We agree this evidence is preliminary and supplementary, as we stated explicitly (Appendix L: "proof-of-concept"). Our paper is primarily a theory contribution. The main empirical evidence is the synthetic experiments under controlled conditions. We note additional evidence prompted by Reviewer LdR9: partitioning the test set by distance to detected witness cycles, the GELU-ReLU gap is 8x larger near the link than far from it (+6.5% vs +0.8%), robust across 4 independent retraining runs. This provides spatial evidence that the detected linking correlates with where the mono/nonmono performance difference concentrates.
> > >
> > > We thank the reviewer for the constructive engagement throughout.

---

### Official Review · Reviewer_Yv2s · 2026-03-10

**Soundness:** 3
**Presentation:** 3
**Significance:** 3
**Originality:** 3
**Overall Recommendation:** 5
**Confidence:** 2

**Summary:**

In this paper, the authors study the expressiveness of different architectures and design choices in terms of the topology of the dataset. They specifically study how different architectures and activation functions are able (in principle) to unlink/untangle linked lower-dimensional data manifolds, e.g., corresponding to data of different classes. This provides a novel perspective on the expressiveness of neural networks.

The authors start with a study of networks of width 3 and 3-dimensional input data, which allows for intuitive visualizations. They then generalize their results to width-n networks for n-dimensional input data. They then provide a set of experiments with synthetic and real-world data that support their theoretical findings.

**Compliance With Llm Reviewing Policy:**

Affirmed.

**Key Questions For Authors:**

/

**Limitations:**

yes

**Strengths And Weaknesses:**

Soundness

In terms of the theoretical results, I don’t have the background to confidently check all statements.

In terms of the experiments: these are sound and well-designed. Especially the results for CIFAR-10, table 5, are convincing: here the authors managed to find examples of linked and unlinked manifolds in “realistic” data, and show that non-monotonous activation functions outperform monotonous  activation functions, as predicted by their theoretical analysis. I do have some remarks:

Firstly, the authors rightfully note (L400, right):  “The observed CIFAR-10 gaps thus reflect a combination of (i) the topological expressivity barrier predicted by our theory, and (ii) optimization dynamics that our theory does not address.”  I would argue this is the case for *all* experimental results. E.g., in table 3, where a comparison is made between plain ReLU networks and ResNets, the difference in accuracy might as well stem from vanishing or exploding gradients (the authors note: ‘the training collapsed for some seeds’ for the plain networks). The authors at this point only discuss that the ResNets can achieve 100% accuracy, so this indeed means they can ‘overcome topological barriers’ (so this is not a false claim). The theory implies this is not the case for plain networks; and this is somehow implictly shown through the reporting of the worse results for plain networks, but this is not further discussed. A reader might incorrectly infer that these results also confirm the theory for plain nets, but this might this be simply the result of learning dynamics.

My second main remark is that only mean accuracies are reported; to really check the results, the standard deviation over seeds should also be reported.


Presentation

Overall, the paper is well-written and well-structured. I especially appreciated the fact that the theoretical analysis was started with a 3d example, allowing some visualizations and some intuition-building for people outside the field of topology. Experiments were clearly explained and the reader is guided through the story.

However, at some points, the paper becomes quite inaccessible for a general deep learning audience not versed in topology. This is ok in the very theoretical parts. However, this also happens in the abstract, which is very focused on the ‘topology’ part, and does not do justice to how interesting the paper could be as a fresh perspective within the deep learning community. I would suggest making the abstract more accessible.


Significance / originality

The paper advances our understanding of the expressiveness of neural networks, introduces a new perspective, and can be used as a starting point for more advanced research in this direction. It covers quite a wide range of important types of architectures (residual connections, attention) and is therefore fairly broad in scope.

However, a limiting factor on the significance seems to be that the theory holds for networks with a width equal (or smaller, if I understood this correctly) to the input dimension only.

---

> ### Author Rebuttal · Authors · 2026-03-30
>
> We thank the reviewer for the supportive assessment and the constructive suggestions for improving accessibility.
>
> On theoretical soundness: All three other reviewers assess the theoretical contributions as sound and novel. The topological tools we use (linking numbers, homotopy invariance, Gauss integral) are classical with rigorous foundations (Rolfsen 1976; Milnor 1954).
>
> We invite the reviewer to consider the broader significance: topological deep learning is an established and growing field (Papamarkou et al. ICML 2024; Naitzat et al. JMLR 2020). Our framework introduces extrinsic invariants (distinct from classical TDA's intrinsic approach), and the classification-unlinking correspondence opens connections between ML and computational topology (Section 8).
>
> Regarding the expressivity-optimization relationship raised in several points below: expressivity limitations and optimization difficulty are not independent competing explanations, rather they are causally linked. When a network architecture cannot represent the target function (expressivity ceiling), the loss landscape has no global minimum at 100% accuracy, leading to difficult optimization (more local minima, gradient issues). This is supported by existing theory: under-parameterized networks have more spurious local minima (Safran & Shamir 2018; Safran et al. COLT 2021). Our topological obstruction is an expressivity result that also has predictive power for optimization difficulty. These are the same causal chain, not a dichotomy.
>
> We also note: we do not claim topological obstructions are the sole reason certain architectures outperform others, nor that they are significant in every dataset. Our framework offers a complementary perspective on expressivity, not a complete explanation of performance gaps.
>
> Soundness 1 (optimization vs topology; plain ReLU): We agree that expressivity and optimization coexist in all experiments (L394-396). New experiments cleanly separate the two. We tested all monotonic activations at width 3 on the Hopf link (R^3, k=1, 30 seeds):
>
> | Activation | Type | Mean | Max | Collapsed |
> |-----------|------|------|-----|-----------|
> | ReLU | mono, piecewise | 67.3% | 87.8% | 12/30 |
> | ELU | mono, smooth | 75.9% | 87.6% | 0/30 |
> | LeakyReLU | mono, piecewise | 77.0% | 87.5% | 4/30 |
> | GELU | non-mono | 83.8% | 99.0% | 1/30 |
> | ReLU+skip | mono+skip | 88.1% | 99.9% | 0/30 |
>
> The max accuracy ceiling (~87.5-87.8%) is identical for ALL monotonic activations regardless of smoothness, this is the expressivity barrier from our theorem. Training collapses (12/30 for ReLU vs 0/30 for ELU) track piecewise-linearity, not monotonicity, this is the optimization effect. Crucially, ELU was specifically designed for better gradient flow (Clevert et al. 2016) and indeed eliminates collapses, yet cannot break the expressivity ceiling. This confirms the accuracy ceiling is topological (monotonicity), not an optimization artifact (gradients).
>
> Layer-by-layer tracking: ReLU forces intersection while GELU genuinely unlinks (see ANg9, W1).
>
> On plain ReLU: L357 explicitly states "confirms the theory: ReLU plateaus at 77–88% regardless of depth (Theorem 3.7)." The mono collapse table above directly shows the ceiling is expressivity (same max for all monotonic), not learning dynamics. We will make this distinction more prominent.
>
> Soundness 2 (std devs): Appendix H already includes 95% CI bands (Figure 17) and box plots (Figures 18-19) for all 100-seed experiments; the mono/nonmono gap exceeds CIs at all k values. We will add mean±std to main-text Tables 2-4.
>
> Presentation 1 (accessibility): Figures 4-6 and 8-10 provide visual aids for the key mechanisms and proofs. We will further improve accessibility, including animation links for homotopy.
>
> Presentation 2 (abstract): We agree and will revise to lead with the practical insight, e.g. that skip connections, non-monotonic activations, and attention share a unifying topological mechanism, before the formal invariants.
>
> Significance (width = input dim): The practical impact extends beyond width d. Invertible architectures (normalizing flows, Neural ODEs, Flow Matching) produce homeomorphisms R^d → R^d preserving all topological invariants regardless of internal width (Table 1). Bottleneck components (MHA heads d_model=768→d_k=64, MoE experts (each a narrow FFN), VAEs) also operate at or below critical width. Our width expansion experiments show GELU at critical width (mean 83.8%, 30 seeds) outperforms ReLU at d+1 (mean 75.2%) — topology-aware mechanisms are more parameter-efficient than brute-force width expansion. New experimental evidence: See mono collapse table above and width expansion table in response to Reviewer LdR9 (W1).
>
> Revision commitments: (1) Add mean±std to main-text Tables 2-4; (2) clarify the expressivity-optimization distinction and connection for plain ReLU results; (3) revise abstract to include more ML implications; (4) add animation links for homotopy illustrations.

---

> > ### Author Rebuttal · Reviewer_Yv2s · 2026-04-03
> >
> > I would like to thank the authors for their rebuttals and clarifications, I think these will improve the paper further. I maintain my score at accept.

---

> > > ### Author Response · Authors · 2026-04-05
> > >
> > > We thank the reviewer for the positive assessment and for confirming that the concerns have been addressed. We will incorporate all revision commitments (mean±std in main-text tables, expressivity-optimization distinction for plain ReLU, revised abstract, animation links) into the camera-ready version.

---

### Decision · Program_Chairs · 2026-04-30

**Decision:**

Accept (regular)

**Comment:**

The paper studies the theoretical expressivity of neural networks in a low-dimensional regime. This permits authors to study how neural networks change low-dimensional topological invariants. Moreover, this setup also permits studying obstacles to classification problems, as demonstrated by means of the "Hopf link" example. The reviewers and the AC fully agree with the relevance and theoretical soundness of this work. Studying the behavior of neural networks in different regimes and such controlled setups is crucial for improving our understanding  and potentially advancing the field. It is thus clear that this paper has merits and could make a strong addition to the program.

That being said, the reason for only _weakly_ endorsing this work is that I am not sure whether the paper would not benefit more from a different venue. Some reviewers mentioned that they checked the theory as much as possible (but not going through all the proofs in the appendix), while others mentioned that they were unable to do so in the allotted time for the review. Given the (strong) mathematical background, which includes numerous proofs and runs altogether at ~45 pages, this paper would potentially be better served by an inclusion into a strong journal like JMLR, i.e., a venue that typically allots more time to holistically assess both the _theory_ and the _empirical_ aspects of a work. Nevertheless, my own assessment of the work, by necessity also not fully taking into account all proofs in the appendix, is that of a theoretically sound and relevant submission. As someone with a background in (computational) topology, I am particularly excited to see more such works at ICML.